# Efficient oral vaccination by bioengineering virus-like particles with protozoan surface proteins

Marianela C. Serradell[1], Lucía L. Rupil [1,2], Román A. Martino[1,9], César G. Prucca[1,9], Pedro G. Carranza[1,10], Alicia Saura[1,2], Elmer A. Fernández[1], Pablo R. Gargantini[1], Albano H. Tenaglia [1], Juan P. Petiti[3], Renata R. Tonelli[4], Nicolás Reinoso-Vizcaino[5], José Echenique [5], Luciana Berod [6,7], Eliane Piaggio[8,11], Bertrand Bellier[8], Tim Sparwasser[6,7], David Klatzmann[8] & Hugo D. Luján[1,2]

Intestinal and free-living protozoa, such as *Giardia lamblia*, express a dense coat of variant-specific surface proteins (VSPs) on trophozoites that protects the parasite inside the host's intestine. Here we show that VSPs not only are resistant to proteolytic digestion and extreme pH and temperatures but also stimulate host innate immune responses in a TLR-4 dependent manner. We show that these properties can be exploited to both protect and adjuvant vaccine antigens for oral administration. Chimeric Virus-like Particles (VLPs) decorated with VSPs and expressing model surface antigens, such as influenza virus hemagglutinin (HA) and neuraminidase (NA), are protected from degradation and activate antigen presenting cells in vitro. Orally administered VSP-pseudotyped VLPs, but not plain VLPs, generate robust immune responses that protect mice from influenza infection and HA-expressing tumors. This versatile vaccine platform has the attributes to meet the ultimate challenge of generating safe, stable and efficient oral vaccines.

[1] Centro de Investigación y Desarrollo en Inmunología y Enfermedades Infecciosas (CIDIE), Consejo Nacional de Investigaciones Científicas y Técnicas (CONICET)/Universidad Católica de Córdoba (UCC), Córdoba X5016DHK, Argentina. [2] Facultad de Ciencias de la Salud, Universidad Católica de Córdoba (UCC), Córdoba X5004ASK, Argentina. [3] Instituto de Investigaciones en Ciencias de la Salud (INICSA), Centro de Microscopía Electrónica, Facultad de Ciencias Médicas, CONICET/Universidad Nacional de Córdoba, Córdoba X5000, Argentina. [4] Departamento de Ciências Farmacêuticas, Instituto de Ciências Ambientais, Químicas e Farmacêuticas, Universidade Federal de São Paulo, Diadema CEP 09913-030, Brazil. [5] Centro de Investigaciones en Bioquímica Clínica e Inmunología (CIBICI), Departamento de Bioquímica Clínica, Facultad de Ciencias Químicas, CONICET/Universidad Nacional de Córdoba, Córdoba X5000HUA, Argentina. [6] Institute of Infection Immunology, Twincore, Centre for Experimental and Clinical Infection Research, Hannover Medical School and Helmholtz Centre for Infection Research, Hannover 30625, Germany. [7] Department of Medical Microbiology and Hygiene, University Medical Center of Mainz, Obere Zahlbacherstr 6755131 Mainz, Germany. [8] Sorbonne Université, INSERM, Immunology-Immunopathology-Immunotherapy (i3), AP-HP, Hôpital Pitié-Salpêtrière, Biotherapy (CIC-BTi) and Inflammation-Immunopathology-Biotherapy Department (i2B), 75005 Paris, France. [9] Present address: Centro de Investigaciones en Química Biológica de Córdoba (CIQUBIC), Departamento de Química Biológica, Facultad de Ciencias Químicas, CONICET/Universidad Nacional de Córdoba, Córdoba X5000HUA, Argentina. [10] Present address: Centro de Investigaciones y Transferencia de Santiago del Estero (CITSE), Facultad de Ciencias Médicas, CONICET/Universidad Nacional de Santiago del Estero, Santiago del Estero CP 4200, Argentina. [11] Present address: Institut Curie, PSL Research University, INSERM U932, Paris 75248, France. These authors contributed equally: Marianela C. Serradell, Lucía L. Rupil. These authors jointly supervised this work: David Klatzmann, Hugo D. Luján. Correspondence and requests for materials should be addressed to H.D.Lán. (email: hlujan@ucc.edu.ar)

Successful vaccination against infectious diseases is considered one of the major accomplishments of medical sciences in history[1]. However, the current trend of vaccine refusal and hesitancy, in part due to perceived potential hazards of parenteral vaccines and the aversion to needles, calls for the development of more friendly and efficient vaccines[2,3]. Mucosal vaccines provide painless and safe administration[4]. While the oral route would be the most convenient mucosal vaccination approach, the nature of the digestive system causes degradation of the ingested antigen by the low gastric pH and digestive enzymes of the small intestine[5]. Consequently, for oral immunization to be effective, protein antigens must be protected from the harsh environment of the upper gastrointestinal tract (GIT).

The early-branching eukaryote *Giardia lamblia* is perhaps the only protozoan capable of colonizing the lumen of the upper small intestine of many vertebrates, including humans[6]. *Giardia* has a simple life cycle consisting of the intestinal flagellated trophozoite and the environmentally resistant cyst[6]. How *Giardia* trophozoites can survive in the hostile milieu of the small intestine is unclear, but the *Giardia* surface is completely covered with molecules belonging to a family of cysteine-rich proteins called variant-specific surface proteins (VSPs)[7–9]. To evade the host immune responses, *Giardia* undergoes antigenic variation by continuously switching its VSPs generated from a repertoire of ~200 homologous genes present in the parasite genome[7,9,10]. Previously, we reported that a mechanism similar to RNA-interference (RNAi) ensures that only one VSP is expressed on the surface of *Giardia* at any time[11]. By knocking down key enzymes of the RNAi pathway, we generated trophozoites expressing their entire repertoire of VSPs[11,12]. Importantly, adjuvant-free oral administration of native VSPs purified from these altered trophozoites afforded efficient vaccination against *Giardia* without causing any symptoms of giardiasis[12,13]. This result indicated that VSPs remain stable and immunogenic after passage through the GIT environment and that they are not toxic to cells or animals[12,13].

VSPs are integral membrane proteins consisting of an extracellular variable region rich in cysteine (mainly as CXXC motifs), a single hydrophobic transmembrane domain and a highly conserved cytoplasmic tail[10]. The molecular mass of *Giardia* VSPs varies from 20 to 200 kDa and the number of CXXC motifs depends on the length of the VSP extracellular region[10]. Surface proteins with the VSP signature (protein family database PF03302) are also present in other parasitic protozoa such as *Entamoeba histolytica* and *E. dispar*, which colonize the large intestine[14], and in the free-living ciliates *Paramecium tetraurelia* and *Tetrahymena thermophila*[15,16]. Therefore, *Giardia* VSPs, or molecules sharing similar characteristics, could be responsible for protecting cells under stress conditions.

It is well known that the most successful vaccines are attenuated or inactivated pathogen-based formulations; i.e., naturally occurring particles[1]. The importance of the particulate form of antigens for efficient vaccination was highlighted by the success of recombinant vaccines based on non-infectious virus-like particles (VLPs)[17,18]. Retrovirus-derived VLPs offer a very versatile and efficient platform for vaccine formulation[19]. We previously showed that many heterologous antigens can be addressed at the surface of these VLPs by fusing their extracellular region with the transmembrane domain and the cytoplasmic tail of the G protein of the vesicular stomatitis virus (VSV-G)[19–21].

Based on these previous findings, we hypothesized that protecting VLPs with VSPs could enable their use for efficient oral vaccination. To test this idea, antigens of the influenza virus, which enter the body through mucosal surfaces of the respiratory tract, were used as model antigens. Influenza are enveloped viruses in which hemagglutinin (HA) is responsible for virus binding to sialic acid-containing molecules, being the main target of neutralizing antibodies (NAbs) that protect against infection[22]. HA can be efficiently pseudotyped onto retrovirus-derived VLPs and its co-expression with neuraminidase (NA) allows efficient VLP release[20,21].

We here show that different VSPs are resistant to proteolytic digestion, fluctuations in pH and temperature and that they have an intrinsic adjuvant activity. When influenza antigens are included in VSP-pseudotyped VLPs, they produce a remarkable immune response against the flu antigens. Oral vaccination with those VLPs protects mice from live influenza virus challenges and from the development of tumors expressing the vaccinal antigen. These results demonstrate that by taking advantage of the properties of surface molecules of protozoan microorganisms oral vaccines can generate protective humoral and cellular immunity locally and at distant sites of the body.

## Results

**Surface proteins containing CXXC motifs are highly resistant**. Regions of different *Giardia* VSPs and VSP-like molecules from *E. histolytica*, *T. thermophila*, and *P. tetraurelia* are shown in Supplementary Fig. 1. The only characteristic common to these cysteine-rich domains is the presence of multiple CXXC motifs, which have been involved in metal-binding[23–25], making intra- and intermolecular disulfide bonds[26] and protecting cells from redox damage[27,28]. Interestingly, all these protozoa were capable of resisting high protease concentrations and remained viable. Conversely, mammalian cells suffered marked morphological alterations and destruction under the same conditions (Fig. 1a and Supplementary Fig. 1). The presence of these proteins on the surface of microorganisms living in harsh environments, characterized by changes of pH, temperature and redox potential as well as the presence of proteolytic enzymes, suggests that surface proteins containing multiple CXXC motifs play a key role in protecting cells under hostile conditions.

To delve into the biochemical properties of VSPs, recombinant extracellular regions of VSPs from three different *Giardia* clones derived from different isolates were produced as soluble proteins in insect cells. The extracellular cysteine-rich region of VSPs was preserved and the C-terminal transmembrane region and the cytoplasmic residues were replaced by a His$_6$ purification tag (ΔVSPs). We first assessed the in vitro behavior of ΔVSP1267 against conditions present in the GIT and observed its high resistance to extreme pH and proteolytic digestion by trypsin, intestinal and stomach extracts (Fig. 1b). ΔVSP9B10 and ΔVSPH7 showed a similar behavior (Fig. 1c). This resistance to proteolysis was identical to that observed in native VSPs from *Giardia* trophozoites (Fig. 1d). In contrast, heterologous antigens such the influenza virus surface protein HA[22] (Fig. 1b) and the *Giardia* intracellular molecule GRP78/BiP[29] (Fig. 1d) showed high susceptibility to degradation, suggesting that the resistance observed in VSPs may depend on the presence of multiple CXXC motifs. For that reason, ΔVSP1267 was treated with either reducing agents or metal chelators and subsequently confronted to trypsin. In these cases, the VSP became sensitive to degradation, which was reversed by metal addition (Fig. 1e), confirming that inter- and intra-molecular cross-linking by metal coordination and/or disulfide bonds are crucial to provide stability to VSPs[23–26].

**CXXC-rich surface proteins activate innate immune cells**. Previous findings showed that *Giardia* VSPs were naturally immunogenic[7,12,13]. Therefore, their potential immunostimulatory properties were analyzed. Signaling through toll-like receptors (TLRs) is a key trigger of immune activation by infectious

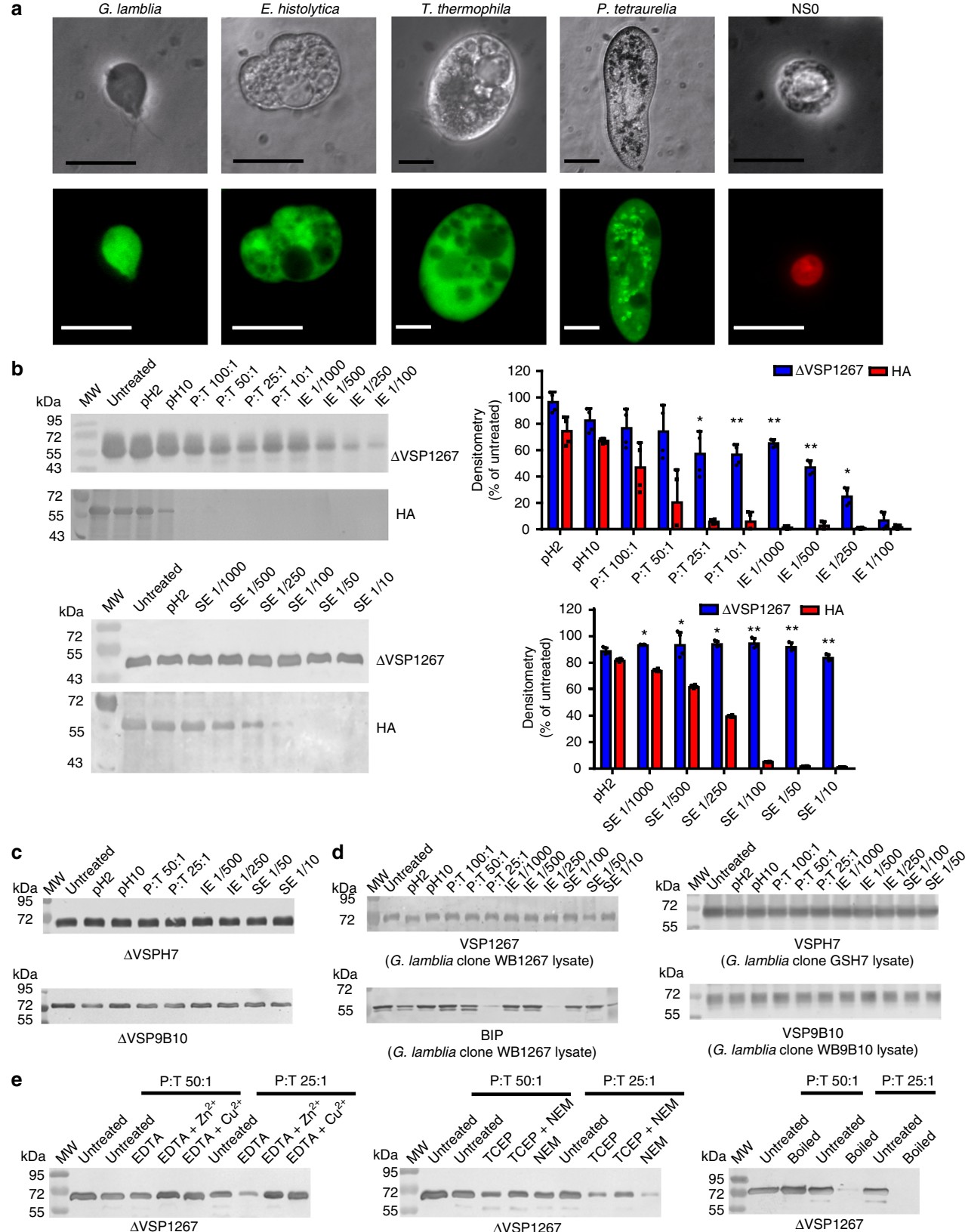

agents[30]. Using a panel of reporter HEK293 cells constitutively expressing human TLRs, ΔVSP1267 was able to signal through TLR-4 and, to a lesser extent, through TLR-2 (Fig. 2a). Additional studies using mouse receptors demonstrated that this activation was dose-dependent and quite specific for TLR-4 (Fig. 2b).

ΔVSP9B10 and ΔVSPH7 also showed a strong and dose-dependent activation of TLR-4. Conversely, recombinant HA expressed and purified using the same approach did not show activation of any of these receptors (Fig. 2b). To rule out any contamination with LPS, which is a known agonist of TLR-4,

**Fig. 1** Resistance to degradation of protozoan CXXC-rich proteins. **a** High magnification representative images of trophozoites from *Giardia lamblia*, *Entamoeba histolytica*, *Tetrahymena thermophila*, and *Paramecium tetraurelia*, and non-adherent mammalian cell (NS0) incubated for 90 min with high trypsin concentration (20 mg ml$^{-1}$). The top panel shows phase contrast images. Bottom panel shows live (green cytoplasm) and dead (red nucleus) images of the same cells stained with fluorescein diacetate and propidium iodide. The bars represent 25 µm. **b**, **c** Recombinant proteins were expressed in insect cells and highly purified by affinity chromatography. Western blotting analysis of the effects of extreme pH, trypsin (T), intestinal extract (IE), and stomach extract (SE) on recombinant proteins. Proteolytic profile of ΔVSP1267 and HA. Representative images are on the left; densitometric measurements are on the right (mean ± s.e.m.) (**b**). Proteolytic profile of recombinant ΔVSPH7 and ΔVSP9B10 (**c**). **d** Proteolytic profile of native VSP1267 compared to an unrelated parasite protein, GRP78/BIP and native VSPs from *Giardia* lysates. **e** Trypsin digestion of ΔVSP1267 subjected to different pre-treatments to modify its structure. The ratio protein:trypsin (P:T) is expressed as w-w. Dilutions of IE and SE are indicated on top. *$p < 0.05$, **$p < 0.01$; Student's *t*-test, $n = 4$ from two independent experiments. Source data are provided as a Source Data file

this compound was quantified in all samples and parallel assays in the presence of polymyxin B were performed. Both strategies showed the absence of detectable bacterial compound in these protein preparations. These results were confirmed by analyzing the activation of bone marrow-derived dendritic cells (BMDCs). ΔVSP1267 induced a significant up-regulation of the expression of CD40 and CD86 co-stimulatory molecules and this activation was not observed when BMDCs from TLR-4 KO mice were used (Fig. 2c), suggesting that multiple CXXC motifs represent a novel pathogen-associated molecular pattern (PAMP) recognized by the pattern recognition receptor (PRR) TLR-4[31].

**VSP-VLPs protect antigens from degradation and enhance their immunogenicity**. The expression of murine leukemia virus (MLV) capsid protein Gag in eukaryotic cells suffices to generate genome free retroviral particles. These particles can be pseudo-typed by various type of envelope proteins or by fusing peptides to the transmembrane domain and cytoplasmic tail of VSV-G[19,20]. We thus engineered VLPs pseudotyped with HA and NA from influenza A H5N1 or H1N1, with or without the co-expression of the extracellular region of VSPs fused to the transmembrane domain and cytoplasmic tail of VSV-G (VLP-HA/VSP-G and VLP-HA, respectively). In some cases, the enhanced yellow fluorescent protein (eYFP) was fused to Gag to obtain fluorescent particles (schematic representation is shown in Supplementary Fig. 2). The expression of each VLP constituent was analyzed by immunofluorescence and transmission electron microscopy (TEM) and the correct assembly and antigen expression of VLPs was determined by western blotting and hemagglutination assays (Fig. 3a–c). Particles showed an average size of 137 nm (VLP-HA) and 165 nm (VLP-HA/VSP-G), according to nanoparticle tracking analysis and electron microscopy (Table 1 and Fig. 3d, e). The surface of the VSP-pseudotyped VLPs looked like the surface of *Giardia* trophozoites, showing the same dense coat that protects *Giardia* from degradation within the gut[8].

Therefore, the resistance to proteolysis of molecules displayed onto VLPs was evaluated. These chimeric VSPs with heterologous transmembrane and cytoplasmic domains displayed similar proteolysis resistance to that of the native ones (Fig. 4a). Most importantly, HA from three VLP-HA/VSP-G expressing different VSPs was more resistant to proteolysis than HA from VLP-HA, showing that VSPs can protect heterologous molecules from degradation (Fig. 4a).

As observed with the recombinant proteins, VLP-HA/VSP1267-G but not VLP-HA induced the activation of TLR-4 reporter cells (Fig. 4b). Additionally, VSP pseudotyping enhanced binding and uptake of VLPs by BMDCs (Fig. 4c) and triggered expression of CD40 and CD86 (Fig. 4d), and increased the release of TNF-α, IL-10, and IL-6 when compared to VLP-HA (Fig. 4e). These effects were abolished when DCs from TLR-4 KO mice were used (Fig. 4d, e), indicating that the presence of VSPs on VLPs improved VLP immunogenicity in a TLR-4 dependent manner.

Importantly, VLP-HA/VSP-G particles were resistant to freezing and thawing and remained stable at different temperatures for over one month without losing their immune-enhancing properties (Fig. 4f).

**VSP-VLPs orally administered induce robust immune responses to HA**. To determine whether the properties of VSPs observed in vitro would translate into efficient mucosal vaccination; mice were immunized by orogastric administration of different VLP formulations (four weekly doses). VLP-HA or VLP-HA/VSP1267-G were given to the animals and the local and systemic anti-HA immune responses were analyzed. HA-specific antibody responses were first studied in serum, fecal extracts and bronchoalveolar lavage (BAL). During the immunization period, the animals did not show any signs of discomfort nor significant weight loss. Sera from mice orally vaccinated with VLP-HA had no reactivity against HA. In contrast, VLP-HA/VSP-G vaccinated mice showed detectable levels of serum total IgG anti-HA (Fig. 5a). After four administrations, all mice immunized with VLP-HA/VSP-G showed anti-HA IgG$_1$ and IgG$_{2a}$ responses (Fig. 5b). These mice also showed secretory IgA responses in BAL and feces[32], which were not observed with plain VLP-HA vaccination (Fig. 5c). The antibodies elicited by oral VLP-HA/VSP-G vaccination persisted in serum at high levels for up to 125 days post immunization (Fig. 5d).

To test whether VSPs must be physically associated with HA on the same VLP, a mixture of VLP-HA and VLP-VSP-G was administered orally to mice and the induction of specific antibodies anti-HA was analyzed. When HA was in VLPs separated from VLP expressing VSP-G, no anti-HA antibodies were found in serum (Fig. 5e), indicating that VSP-G must be in the same particle as HA to provide protection to the heterologous antigen.

Given that parentally administered VLPs are highly immunogenic[19,33], we also evaluated the effect of VSP using this route of administration. Similar to oral administration, sub-cutaneously administered HA-VLPs induced specific anti-HA serum antibodies. However, subcutaneous immunization with VLP-HA/VSP-G achieved higher levels of anti-HA IgG than with VLP-HA. This indicates that the presence of VSP onto the VLP not only fulfills the protective effect observed using oral administration but also increases the immunogenicity of systemically administered VLPs (Fig. 5f). Conversely, a significant amount of fecal anti-HA IgA was only observed with oral VLP-HA/VSP-G vaccination (Fig. 5f).

A great number of VLPs have been used as vectors for the delivery of heterologous vaccine antigens. However, for some of these vectors, a pre-existing immunity could reduce the immune response to a vectored antigen[34]. For this reason, VSP1267-pseudotyped VLP-HA was used in mice previously immunized with recombinant ΔVSP1267. Despite the expected presence of anti-VSP antibodies[13], VLP-HA/VSP-G elicited the production of

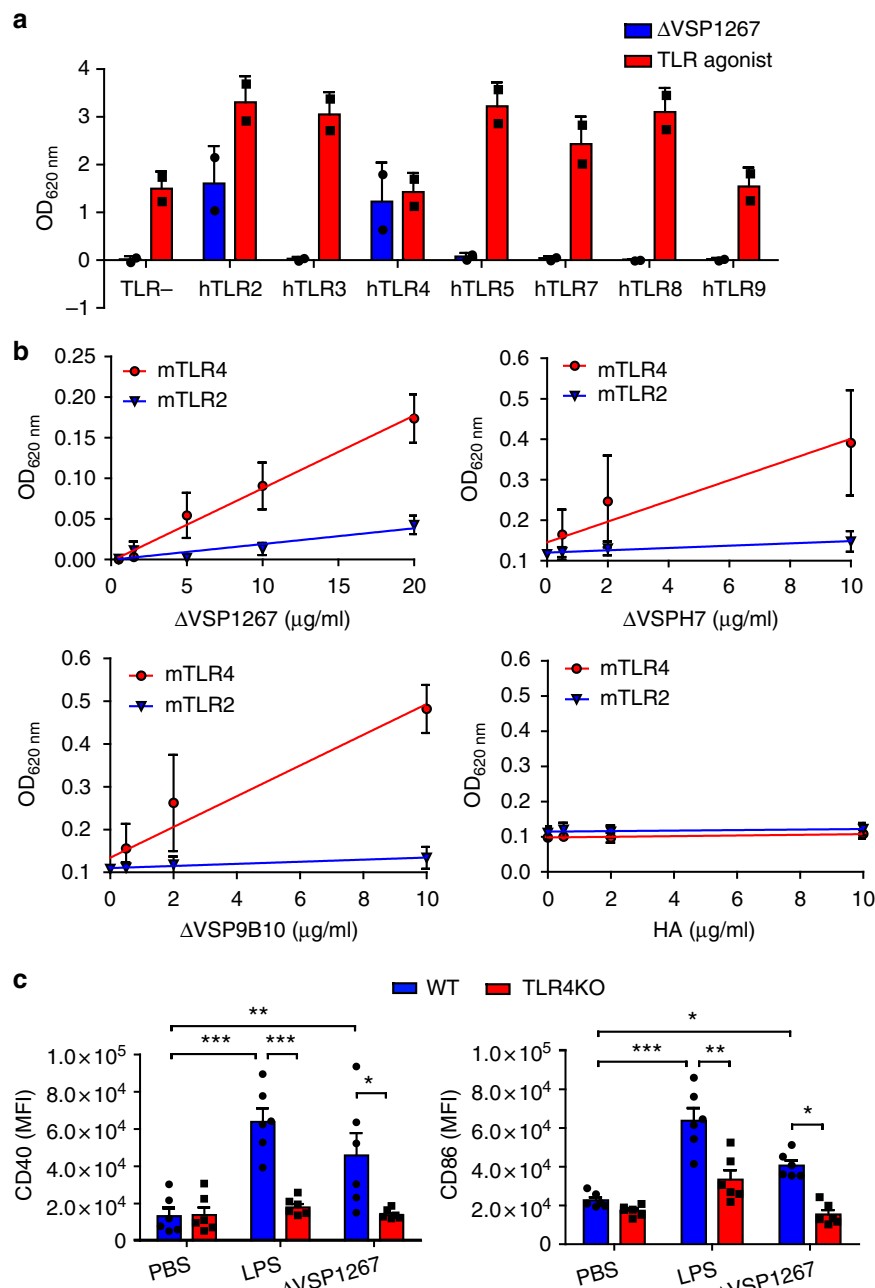

**Fig. 2** Immunogenic properties of different *Giardia* VSPs. **a** Activation of a panel of human TLR reporter cells (hTLR) by ΔVSP1267 (20 μg ml⁻¹). Reporter cells were incubated with the recombinant protein and the secretion of SEAP was recorded. Specific TLR agonists were employed as positive controls (hTLR-2: Pam2CysK4, hTLR-3: PolyI:C, hTLR-4: LPSK12, hTLR-5: flagellin, hTLR-7: R848, hTLR-8: R848, hTLR-9: ODN2006). The parental cell line was used as negative control (TLR-). **b** Activation of mouse TLR-4 and TLR-2 reporter cell lines (mTLR) by ΔVSP1267, ΔVSPH7, ΔVSP9B10, and recombinant HA (H5N1). *$p < 0.05$ ΔVSPH7 and ΔVSP9B10 for TLR-4, **$p < 0.01$ ΔVSP1267 for TLR-4 and TLR-2; Linear regression test, $n = 6$ from three independent experiments. **c** In vitro stimulation of BMDCs. C57BL/6 WT and TLR-4 KO BMDCs were incubated in vitro with PBS, LPS (500 ng ml⁻¹) or ΔVSP1267 (50 μg ml⁻¹), and CD40 and CD86 levels were measured by flow cytometry (MFI: mean fluorescence intensity). *$p < 0.05$, **$p < 0.01$, ***$p < 0.001$; two-way ANOVA, Bonferroni post-tests, $n = 6$ from three independent experiments. Values represent mean ± s.e.m. Source data are provided as a Source Data file

anti-HA serum IgG and fecal IgA at similar levels than in non-pre-immunized mice (Fig. 5g).

Finally, mice vaccinated with VLP-HA/VSP-G also developed a cellular immune response evidenced by HA-specific IFN-γ-producing T cells (Fig. 5h) and a significant increase of TNF-α and IL-6 production by activated splenocytes (Fig. 5i). Thus, VSP pseudotyping was effective in protecting and adjuvanting the

heterologous HA antigen in vivo by triggering systemic and mucosal, cellular and humoral immune responses after oral vaccination.

**VSP-VLPs orally administered induce protective immune responses**. We then investigated the efficacy of these induced humoral and cellular responses against viral and tumor cell challenges, which are mainly controlled by antibody and cellular

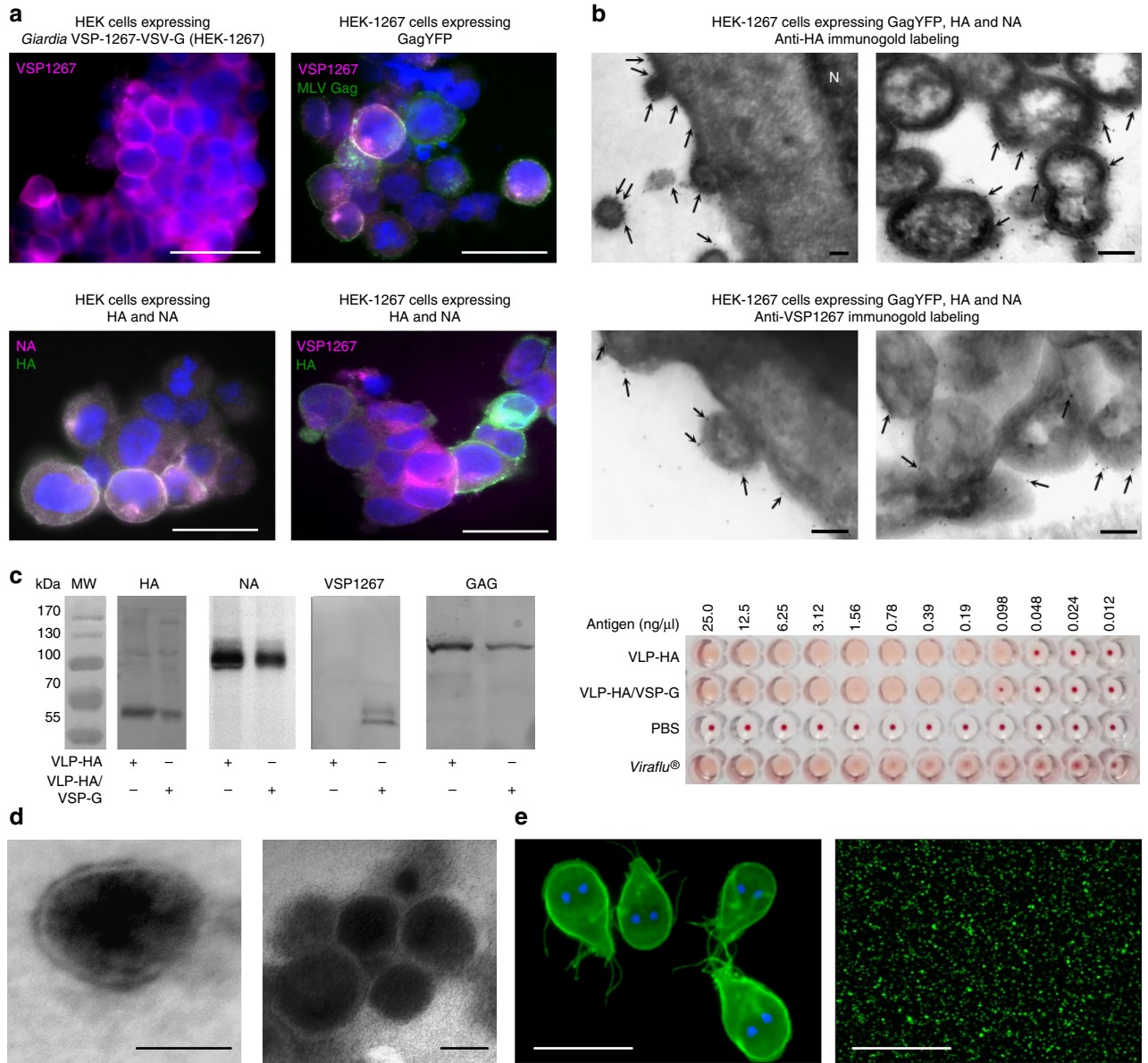

**Fig. 3** VLP characterization. **a** Immunofluorescence microscopy of HEK cells used in VLPs production. The different DNA constructs used are indicated on top of each figure. In all immunofluorescence images, nuclei are labeled with DAPI (blue). The white signal represents colocalization. The bars represent 20 μm. **b** Electron micrographs of 100 nm-thick cryosections showing VLPs budding from the surface of HEK-1267 cells. Five nanometer immunogold labeling of HA (top panels) and VSP1267 (bottom panels) indicates their localization in cells and VLPs membranes (arrows). N, nucleus. The bars represent 100 nm. **c** Western blotting and hemagglutination assays showing the correct assembly of VLPs. **d** Representative TEM negative staining of VLPs (VLP-HA on the left and VLP-HA/VSP1267-G on the right). The bars represent 100 nm. **e** Immunofluorescence images of VSPs on the surface of *Giardia* trophozoites (left) and of VSP-G on the surface of VLPs (right). In both cases, mAb 7F5-FITC anti-VSP1267 extracellular domain was used. VSPs are labeled in green and nuclei stained with DAPI in blue. The bars represent 25 μm. Source data are provided as a Source Data file

**Table 1 Analysis of the size distribution and concentration of VLPs by nanoparticle trace analysis (NTA)**

| Sample | Size (nm) | | | Concentration (particles ml⁻¹) |
|---|---|---|---|---|
| | Mean | Mode | s.d. | |
| VLP-HA | 136.9 | 140.5 | 48.4 | $1.31 \times 10^9$ |
| VLP-HA/VSP-G | 164.7 | 147.4 | 63.9 | $8.27 \times 10^8$ |

VLP virus-like particle, VSP variant-specific surface protein, HA hemagglutinin

responses, respectively. Vaccinated and control mice were infected by the nasal route with a mouse adapted strain of live influenza A H5N1 virus[35]. VLP-HA given orally did not protect the mice from the virus challenge, whereas mice orally immunized with VLP-HA/VSP1267-G were fully protected, indicating the presence of a highly efficient humoral response in the respiratory tract. Control animals receiving recombinant HA plus alum intramuscularly were only partially protected (Fig. 6a), similar to the reported efficacy of the current human vaccines[22].

Challenged mice were monitored daily for disease signs (ruffled fur, dyspnea, lethargy) and body weight changes for 2 weeks (Fig. 6b). No significant body weight loss or clinical signs of infection were observed in non-challenged mice or in mice

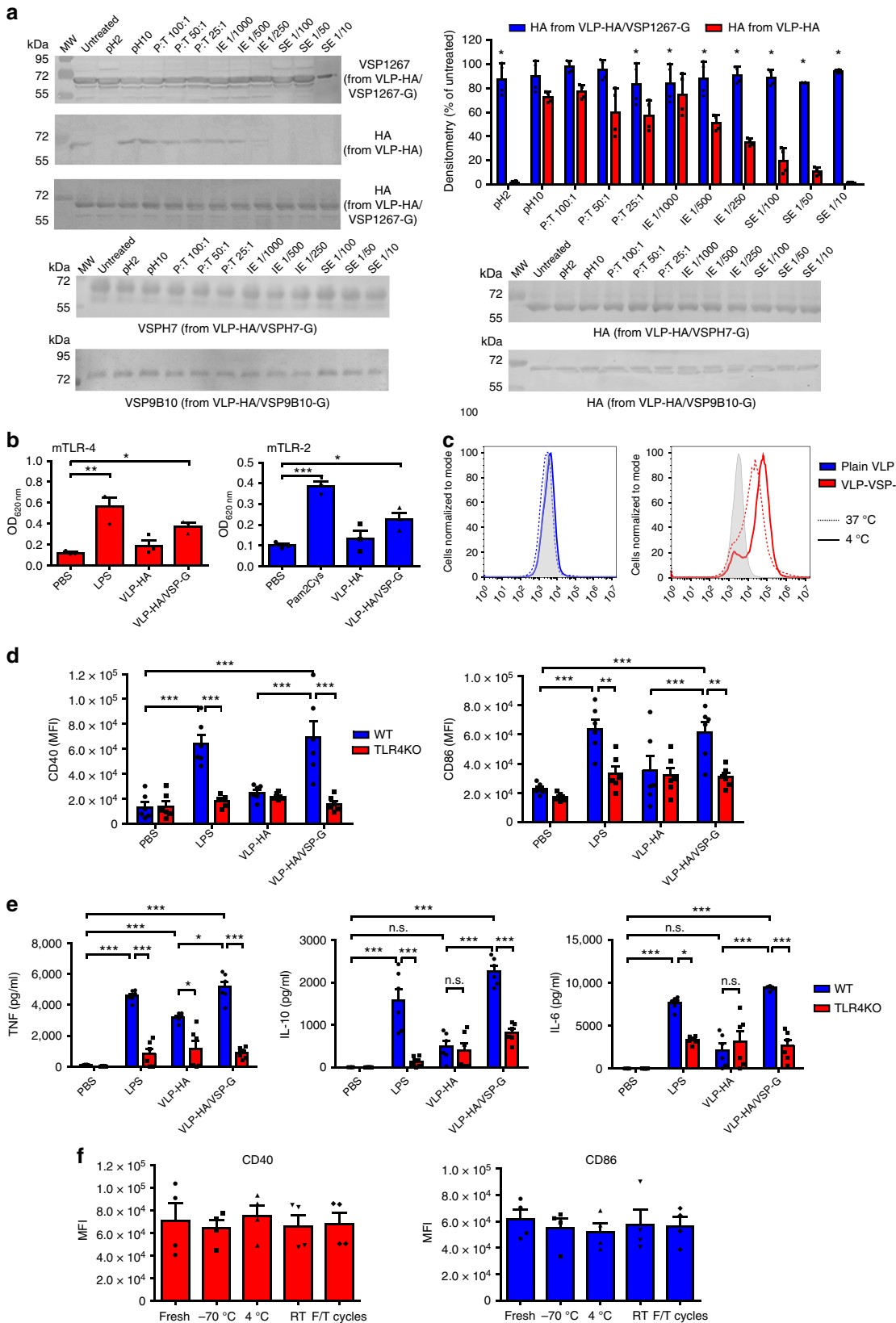

vaccinated with oral VLP-HA/VSP-G. In contrast, unvaccinated mice, mice orally vaccinated with VLP-HA and subcutaneously vaccinated with recombinant HA plus alum rapidly lost weight and died or had to be euthanized at 4 to 6 days post challenge due to severe clinical symptoms.

For the tumor challenge, vaccinated and control mice were injected with the murine AB1 malignant mesothelioma that expresses HA from influenza A H1N1[36,37]. AB1 tumors of large size developed in non-vaccinated mice and in those orally vaccinated with VLP-HA, whereas oral immunization with

**Fig. 4** Resistance to degradation and immunogenic properties of VSP-pseudotyped VLPs. **a** Western blot analysis of the action of extreme pHs, T, IE, and SE on VSP1267, VSPH7, VSP9B10, and HA on the surface of different VLPs. The densitometric analysis shows the relative intensity of HA from VLP-HA in comparison to HA from VLP-HA/VSP1267-G. The ratio protein:trypsin (P:T) is expressed as w-w. Dilutions of IE and SE are indicated on top. $*p < 0.05$; Student 's $t$-test, $n = 4$ from two independent experiments. **b** Activation of mTLR-4 and mTLR-2 reporter cell lines by VLP-HA and VLP-HA/VSP1267-G. $*p < 0.05$, $**p < 0.01$, $***p < 0.001$; one-way ANOVA, Tukey's multiple comparison test, $n = 3$ from three independent experiments. **c** Binding (solid line) and uptake (dotted line) of VLP-VSP-G vs. plain VLP. BMDCs were incubated with fluorescent particles and analyzed by flow cytometry. **d** In vitro activation of WT and TLR-4 KO BMDCs by VLP-HA or VLP-HA/VSP-G. MFI quantification of CD40 and CD86 expression is shown. $**p < 0.01$, $***p < 0.001$; two-way ANOVA, Bonferroni post-tests, $n = 6$ from three independent experiments. **e** Cytokine levels in BMDCs supernatants. Only those cytokines with detectable levels are shown. $*p < 0.05$, $***p < 0.001$; two-way ANOVA, Bonferroni post-tests, $n = 6$ from three independent experiments. **f** BMDCs were incubated with VLP-HA/VSP-G treated as freshly purified (Fresh), ten cycles of freezing and thawing (F/T), or kept at $-70\,°C$, $4\,°C$, or room temperature (RT) for 1 month. CD40 and CD86 levels were measured by flow cytometry. MFI quantification is shown; one-way ANOVA, Tukey's multiple comparison test, $n = 4$ from three independent experiments. Values represent mean ± s.e.m. Source data are provided as a Source Data file

VLP-HA/VSP-G showed an almost complete control of tumor growth (Fig. 6c, d). Antibodies generated by oral vaccination were capable of neutralizing influenza A H1N1 virus from infecting MDCK cells in vitro (Fig. 6e), confirming not only the correct conformation of HA on the surface of the VLPs but also the capability of NAbs to bind to essential epitopes of the viral molecule.

In line with these results, oral vaccination with VSP-pseudotyped particles generated a robust IFN-γ T-cell response (Fig. 6f) and in vitro cytotoxicity against the HA-expressing tumor cells (Fig. 6g). We thus tested whether these responses would be sufficient for therapeutic vaccination of mice that were already developing tumors. Vaccination consisting of four oral doses given every three days after detectable tumor formation did not prevent expansion of the HA-expressing AB1 mesothelioma tumor (Fig. 6h, left panel). In contrast, under the same conditions, therapeutic vaccination remarkably reduced HA-expressing 4T1 murine breast cancer tumor size to undetectable levels (Fig. 6h, right panel). Altogether, the humoral and cellular anti-HA immune responses produced by oral immunization with VLP-HA/VSP-G translated well into protective anti-viral and anti-tumor immune responses.

## Discussion

Achieving efficient oral immunization is considered the ultimate goal of vaccinology. However, despite the introduction of the Sabin oral polio vaccine in the early 1960s, only a few mucosal vaccines have been successfully developed and most of them have been withdrawn from the market[38,39]. One main reason for these failures is the loss of antigen immunogenicity due to the digestive conditions of the GIT[5]. Of the many biotechnological approaches used to protect antigens, encapsulating microparticles and liposomes are among the most used cores for delivering vaccinal antigens by the oral route[40]. However, particle components, including vaccine antigens, are usually destroyed by bile emulsifying factors as well as intestinal and pancreatic secretions containing hydrolytic enzymes released to the lumen of the small intestine with the purpose of favoring food digestion[5,32]. Therefore, we hypothesized that using the strategies developed by intestinal microorganisms to survive under hostile conditions could protect antigens from degradation within the GIT.

*Giardia* colonizes the lumen of the upper small intestine of many vertebrate hosts[6]. In that harsh environment, *Giardia* is protected by a tight coat composed of VSPs[6,8]. Our results demonstrate that VSPs are highly resistant to degradation, likely due to their capability of coordinating metals and forming both intra- and intermolecular disulfide bonds[24,26]. Formerly, it was reported that only certain VSPs were able to protect *Giardia* trophozoites from the action of intestinal proteases and suggested that those differences may be related to VSP host specificity[41].

Our results, however, show that all tested protozoan expressing surface proteins containing multiple CXXC motifs were resistant to high protease concentration and that three different VSPs (VSP1267 and VSP9B10 belonging to assemblage A isolate WB and VSPH7 belonging to assemblage B isolate GS/M) not only were resistant to purified intestinal proteases but also to stomach and intestinal extracts containing a variety of serine-, carboxy- and endo- proteinases[12,13], which also demonstrates that VSPs are not acting as inhibitors of proteolytic enzymes.

Considering the exploitation of VSPs in vaccine design, we also investigated their immune-enhancing properties. Microbial PAMPs are recognized by the innate immune system through PRRs. PRRs on the surface of DCs, M cells and epithelial cells of the intestine are known to be important in antigen transcytosis and presentation[42]. In this work, we observed that *Giardia* VSPs were able to signal through TLR-4 both in DCs and in a reporter cell line. Activation of this receptor has been shown to be immunostimulatory, generating not only an inflammatory response but also the activation of the adaptive arm of the immune system[30]. Thus, proteins containing multiple copies of the CXXC motif can be considered a novel PAMP[31]. This finding might explain the efficacy of the protective response generated by the anti-*Giardia* oral vaccine composed of the entire repertoire of VSPs in experimental and domestic animals such as gerbils, dogs and cats[12,13]. Therefore, VSPs could be used not only to protect antigens from degradation but also to provide a desirable mucosal adjuvant effect.

We investigated the use of VSPs in vaccine design using VLPs as a platform. The most widely used vaccines are based on attenuated or inactivated bacteria or viruses, emphasizing the importance of a particulate structure for proper immunization. Recombinant vaccines are safer than traditional vaccines, but are often less immunogenic and usually require multiple doses and effective adjuvants[43]. VLPs used for homologous vaccination are a highly effective subunit vaccine that mimics the overall structure of virus particles and thus preserves the native antigenic conformation of the immunogenic proteins without containing infectious genetic material[17,18]. VLPs also make excellent carrier molecules for the delivery of heterologous antigens because their particulate structure is readily taken up by antigen-presenting cells, and is thus able to prime long-lasting cytotoxic T lymphocytes (CTL) responses in addition to antibody responses[44,45]. In addition, VLPs based on enveloped viruses, such as retrovirus-based VLPs, have the unique property of being able to express the envelope protein of heterologous viruses under their proper tertiary and even quaternary structures, thus ideal for the triggering of NAbs[18–21].

The ease of designing antigen-presenting VLPs offers a promising approach for the industrial production of vaccines against many diseases. VLPs have been produced in a wide range of taxonomically and structurally distinct viruses, combining unique

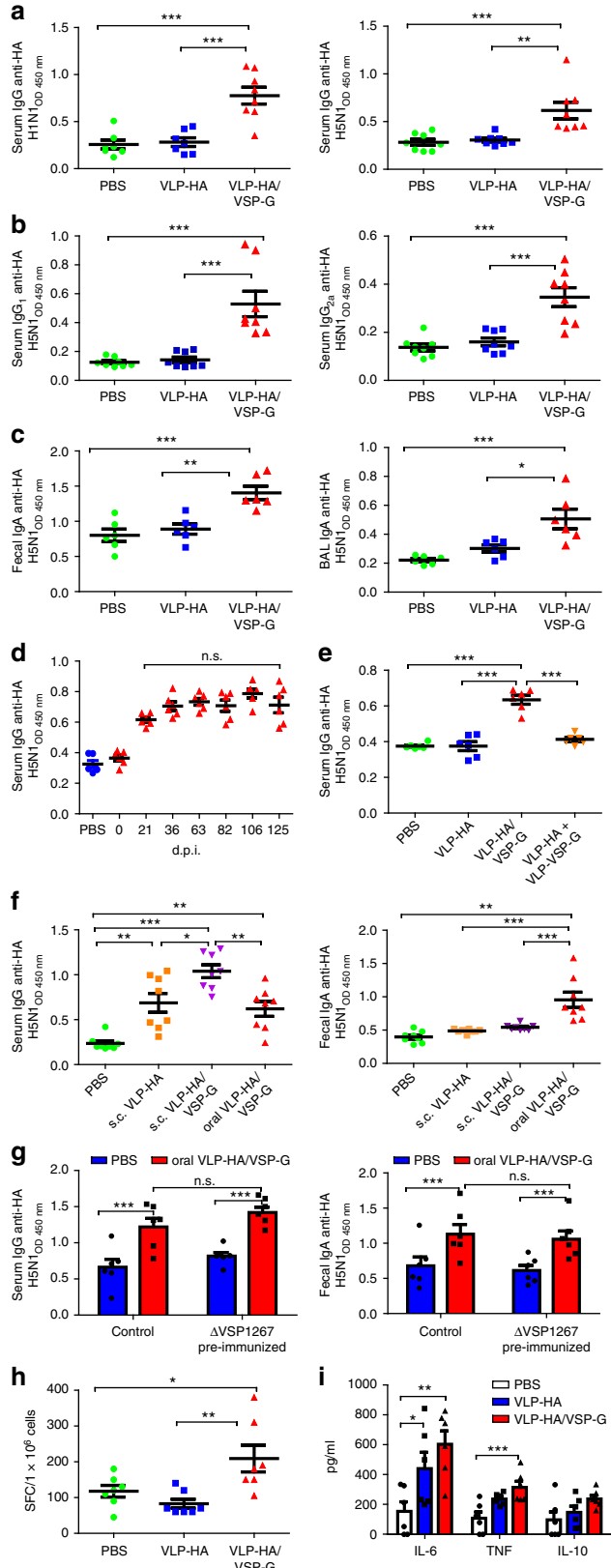

**Fig. 5** Induction of anti-HA humoral and cellular responses in mice vaccinated with VLP-HA/VSP-G. **a–e** Mice were orally vaccinated with VLPs and the humoral response against HA in different biological samples was evaluated. Levels of total serum anti-HA IgG (**a**), IgG1 and IgG2a (**b**), $n = 8$ from two independent experiments. Levels of IgA anti-HA in fecal extracts and BAL, $n = 6$ from two independent experiments (**c**). Levels of serum IgG anti-HA measured at different times post immunization, $n = 6$ from two independent experiments (**d**). Levels of serum IgG anti-HA in animals immunized with a mixture of VLP-HA plus VLP-VSP-G, $n = 6$ from two independent experiments (**e**). **f** Levels of serum IgG and fecal IgA in mice vaccinated with VLPs via oral or subcutaneous (s.c.) route, $n = 8$ from two independent experiments. **g** Mice were orally immunized with four weekly doses of 50 µg of recombinant ΔVSP1267 or vehicle. One week after the last immunization the presence of anti-VSP1267 antibodies in blood and fecal samples was checked, then these animals were orally immunized with VLP-HA/VSP-G, according to the protocol previously used. One week after the last VLP dose, the levels of serum IgG and fecal IgA anti-HA Abs were measured, $n = 6$ from two independent experiments. **h**, **i** Mice were orally vaccinated with VLPs and the cellular response against HA was evaluated. The frequency of HA-specific IFN-γ-secreting T cells (spot-forming colonies, SFC) was determined after antigen-specific re-stimulation of splenocytes, $n = 7$ from two independent experiments (**h**). Cytokines were measured in splenocyte supernatants and only those cytokines with detectable levels are shown, $n = 6$ from two independent experiments (**i**). Data were analyzed by one-way ANOVA and Tukey's multiple comparison test (**a–f**, **h**, **i**) or by two-way ANOVA and Bonferroni post-tests (**g**). Values represent mean ± s.e.m. *$p < 0.05$, **$p < 0.01$, ***$p < 0.001$. Source data are provided as a Source Data file

available on the market have only been delivered parenterally, showing the same disadvantages that any other vaccine given by injection[18]. Therefore, we hypothesized that harnessing VLP vaccines with VSPs could allow their oral administration while providing adjuvant properties. For this purpose, we used retrovirus-based VLPs, which allow efficient generation of NAbs, as demonstrated for a variety of antigens[17–21]. Retrovirus-based VLPs expressing heterologous antigens have already been produced for clinical trials, demonstrating the ability to be manufactured with yields and purity that are expected to be suitable for vaccine production at a commercial scale[21].

As a proof of concept, we used antigens from influenza virus[4]. We designed and produced enveloped chimeric VLPs that co-express VSPs and the main influenza virus antigens on their membrane. Remarkably, VSP expression protected HA from the action of proteolytic enzymes and intestinal and stomach extracts. Previously, protection of membrane molecules in trans was observed only during *Giardia* infections[7,50]. Protection of HA on the surface of VSP-pseudotyped VLPs is the first demonstration that such effect can be generated artificially and translated to other membranes. How VSPs protect heterologous antigens is still an open question. However, we have shown that (a) the heterologous antigens must be present on the same particles that express the VSP-G, (b) NA and HA are exposed on the surface of VLPs since NA is accessible to its substrate and HA retains its hemagglutinating activity, (c) HA and VSP are exposed because they are detected on the surface of VLPs by immunoelectron and immunofluorescence microscopy, and (d) VSPs do not act as protease inhibitors. All these results suggest that the heterologous antigens are not directly shielded by VSP-G molecules on the surface of the VLPs. Therefore, we hypothesize that intra- and intermolecular interactions among these proteins, likely involving metal and disulfide bridges, play crucial roles in antigen protection. Indeed, when VSPs are treated with metal chelators or reducing agents their protective properties are lost. Nevertheless,

advantages in terms of safety and immunogenicity as exemplified by the current vaccines against the Human Papilloma Virus (HPV) and Hepatitis B virus (HBV)[17,18]. On the other hand, although several routes of administration have been used in different vaccination trials with VLPs, including nasal[46], intra-vaginal[33], rectal[47], skin[48], and oral[49] routes, VLP vaccines that are

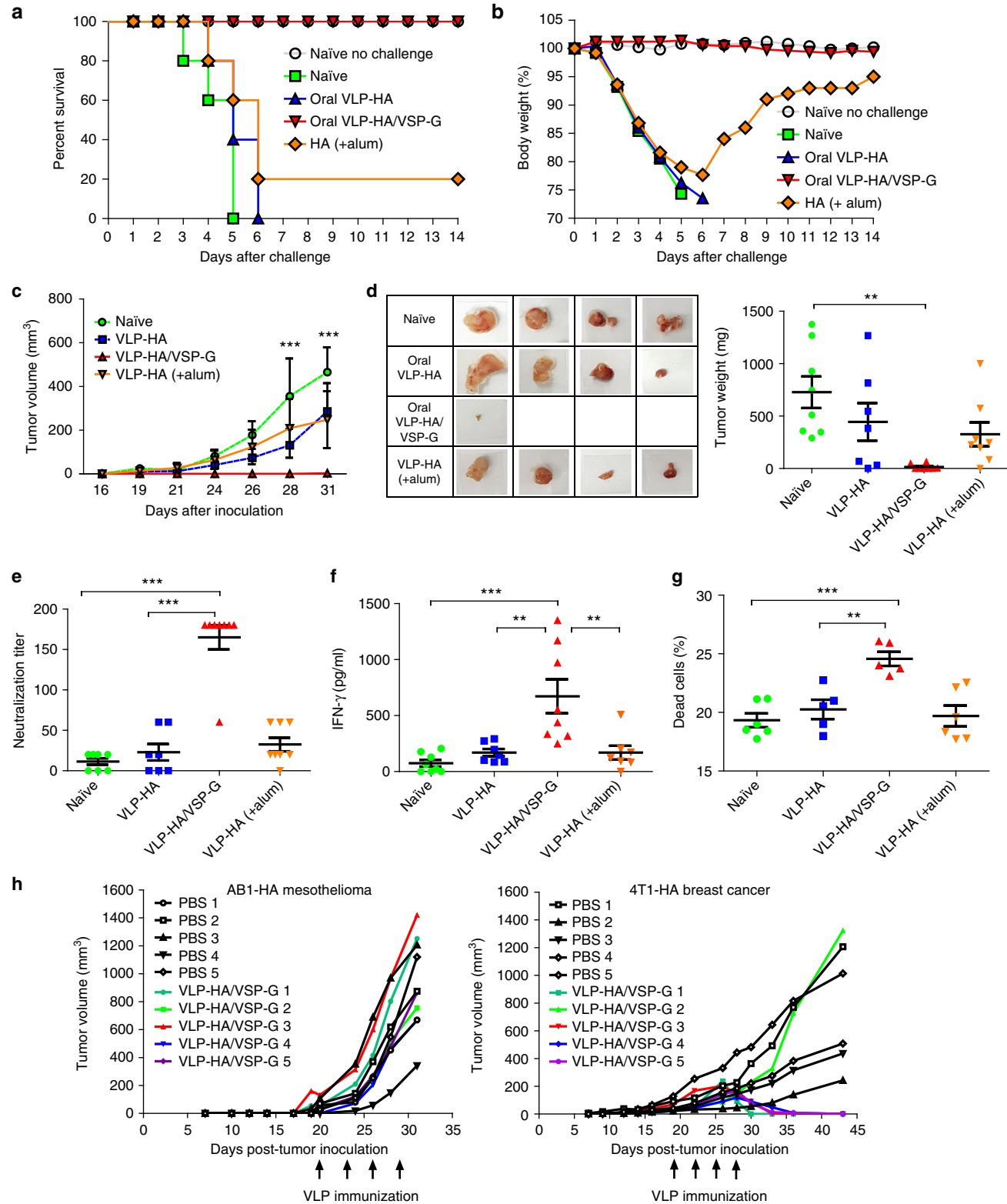

the completion of the high-resolution structures of ΔVSP and ΔVSP-antigen complexes, which has been elusive until now, will allow a better understanding of the mechanism by which VSPs interact and protect heterologous antigens from the action of proteolytic enzymes.

VSPs not only protected HA from degradation but also enhanced the in vitro immunogenic properties of VLP-HA. Particle binding and uptake, up-regulation of co-stimulatory

molecules as well as secretion of the pro-inflammatory cytokines TNF-α and IL-6 by DCs were improved in VSP-pseudotyped VLPs with respect to plain VLPs. Production of cytokines by DCs is crucial in modulating immune responses generated through microbial products[51]. TNF-α is an important pleiotropic cytokine involved in host defense and inflammation[52]. IL-6 is a multi-functional cytokine involved in regulation of immune responses, acute-phase responses, and inflammation[53]. IL-6 promotes T-cell

**Fig. 6** Protective efficacy of VLP-HA/VSP-G oral vaccination against influenza virus challenge and HA-expressing tumors. **a**, **b** Ten days after H5N1 VLP immunization, mice were challenged intranasally with a mouse-adapted influenza virus, $n = 5$. Kaplan–Meier life survival curve analysis was performed using the log-rank Mantel-Cox method for curve comparison analysis (**a**). Body weight is plotted as a percentage of the average initial weight taken at day 0. The body weight changes were evaluated for 2 weeks (**b**). **c**, **d** Mice immunized with H1N1 VLPs were injected 10 days after the last dose with AB1-HA tumor cells. Graph showing the tumor volume growth, $n = 8$ from two independent experiments (**c**). Thirty-one days after tumor inoculation the tumors were harvested and weighed, and representative tumor photographs are shown, $n = 8$ from two independent experiments (**d**). **e** Ten days after H1N1 VLPs immunization, the titer of neutralizing antibodies was measured in sera by a standard microneutralization assay, $n = 8$ from two independent experiments. **f**, **g**, Ten days after H1N1 VLPs immunization, mice were injected with AB1-HA cells and after 31 days the mice were sacrificed. IFN-γ was measured in HA re-stimulated splenocyte supernatants, $n = 8$ from two independent experiments (**f**). In vitro cytotoxicity assay using splenocytes and CFSE-labeled AB1-HA as target cells was performed. Dead cells quantification on CFSE + cells is shown, $n = 6$ from two independent experiments (**g**). **h** AB1-HA or 4T1-HA tumor cells were inoculated (day 0; $n = 10$) and once tumors were detected, half of the mice were therapeutically vaccinated (arrows). Data were analyzed by one-way ANOVA and Tukey's multiple comparison test (**d** to **g**) or by two-way ANOVA and Bonferroni post-tests (**c**). Values represent mean ± s.e.m. **$p < 0.01$, ***$p < 0.001$. Source data are provided as a Source Data file

proliferation, B-cell differentiation and survival, and plasma cell production of IgG, IgA, and IgM[54]. The adjuvant effects of VSP-pseudotyped VLPs were TLR-4 dependent since they were abolished when DCs from TLR-4 KO mice were used. Most likely, intestinal DCs sampling the lumen of the small intestine for particulate antigens are key players in activating a broad immune response[55].

These observed effects in vitro translated well in the capacity to elicit efficient immune responses in vivo. When mice were orally immunized with HA-expressing VLPs, without added adjuvants, a strong cellular and antibody response was generated with VLP-HA/VSP-G, but not with plain VLPs. These results indicate that VSP pseudotyping was necessary to protect and adjuvant VLPs to generate a humoral response that not only is present in serum but also prevails in mucosal fluids, which are the first barrier encountered by infectious agents[38,39,56].

The gut is also a site for mucosal tolerance induction, as occurs with food antigens[57]. While tolerance is important in preventing pathology, it has been considered an obstacle for the development of oral vaccines[57,58]. However, this is not the case with VSP-pseudotyped VLPs, as intact antigens rather than degraded peptides can reach the immune cells of the small intestine and VSPs have strong adjuvant properties.

Most importantly, these cellular and humoral immune responses translated into vaccine efficacy. We used two proto-typic models for assaying humoral and cellular responses, in which the protective immunological mechanisms are essentially very different: protection from live virus infection and killing of tumor cells.

It is widely accepted that an effective anti-influenza vaccine must elicit NAbs. Here, the protection from an intranasal challenge with live influenza virus was indeed correlated with the generation of neutralizing anti-HA antibodies. As the formation of these antibodies is CD4$^+$ helper T-cell dependent, these results also indicate that efficient immune cellular responses were triggered by orally administered VSP-pseudotyped-VLPs[59]. On the other hand, protection from tumor challenge is largely dependent on the generation of CTL, with a possible contribution of antibody-dependent cell-mediated cytotoxicity and the induction of IFN-γ. Here, the protection from AB1 HA-expressing tumor cells observed after oral immunization with the VSP-pseudotyped-VLPs indicates that efficient cytotoxic responses were mounted. A positive correlation was also noted between IFN-γ production and the tumor growth control (Fig. 6d, f) while no such correlation was observed with antibody levels (Figs. 5a and 6d).

VSP-based oral immunization against tumors could possibly be used therapeutically. Although therapeutic oral vaccination was not effective against the mesothelioma tumor, it led to tumor eradication when using the HA-expressing 4T1 murine breast cancer. Several factors could contribute to the observed differential outcome. Among these, the anti-tumor immune response, tumor cell number, growth rate, size, and the particular tumoral microenvironment are likely the most significant[36]. Nevertheless, the 4T1 tumor control/reversion highlights the strength, quality and the effectiveness of the induced cytotoxic immune response elicited by VLP-HA/VSP-G vaccine. It is well known that tumor fate is the uncertain result of a battle between tumor and immune cells in a particular environment[36]. Nevertheless, the unexpected differences between these two HA-expressing tumors make them valuable starting models to better understand the molecular mechanisms underlying the observed effects and the potential manipulation of this vaccine platform to enhance its therapeutic potential against a variety of tumors[36,37].

In the intestine, induction and regulation of mucosal immunity take place primarily in gut-associated lymphoid tissue (GALT)[60]. *G. lamblia* colonizes the upper small intestine where it releases VSPs that are known to interact with the gut epithelial and immune cells[61]. Our results propose that a marked difference takes place between the large intestine and the upper small intestine in determining the outcome of the immune response. Thus, our results would suggest that efficiently targeting the GALT of the upper small intestine, instead of the tolerogenic large intestine, can generate protective immunity rather than tolerance.

Since *Giardia* species are intestinal parasites of many vertebrates, VSPs may have the same protecting and immune-enhancing capabilities for oral vaccination of a variety of animals, including humans. This versatile oral vaccine platform based on VSP-pseudotyped VLPs can be easily adapted to different antigens from infectious agents or tumors and has the attributes to potentially help reduce reticence to vaccination, facilitate mass vaccination programs and be used in remote areas of the world where vaccine refrigeration is impractical.

## Methods

**Animals**. BALB/c, C57BL/6, and C57BL/10ScNJ (TLR-4 KO) mice (6–8 week-old) of both sexes were obtained from the Facultad de Ciencias Veterinarias, Universidad Nacional de la Plata (Argentina) and housed in the vivarium of the CIDIE under specific pathogen-free (SPF) conditions in microisolator cages (Techniplast, Italy); they were cared for following NIH guidelines for laboratory animals. No animals were harmed during the collection of blood and fecal samples. For immunization and challenge studies the group sizes were chosen based on previous experience and littermates of the same sex were randomly assigned to experimental groups. The number of animals for each experiment and all procedures followed the protocols approved by the Institutional Committee for Care and Use of Experimental Animals (CICUAL protocols UCC.2010-36-15p, CIDIE.2016-36-15p-2, and CIDIE.2018-36-15p-3). Tumor volume determinations were carried out in a blinded fashion in different laboratories. All data points were included in the analyses, and no outliers were excluded in calculations of means/statistical significance. The number of samples and experimental replicates are indicated on each figure legend.

**Cell lines**. All cell lines were regularly tested and remained negative for *Mycoplasma spp*. Endotoxin levels in all media and antigens were determined by the *Limulus* amebocyte lysate test (Lonza) using the *Escherichia coli* 0111:B4 internal

control[62]. *Giardia lamblia* trophozoites isolate WB (ATCC® 50803) clones 1267[41] and 9B10[63] and isolate GS/M (ATCC® 50581) clone H7[64] were used. HEK293-1267, HEK293-H7 and HEK293-9B10 cell lines were generated following the instructions of the Flp-In® 293 kit (Invitrogen, Cat. # R75007). They constitutively express on their surface the extracellular portion of a VSP of *G. lamblia* (GenBank: M63966.1/VSP1267; AF293416.1/VSP9B10; M80480.1/VSPH7) fused to the transmembrane domain and the cytoplasmic tail of the VSV-G. HEK293, Flp-In® derivatives, HEK-Blue® TLRs (Invitrogen, Cat. # hkb-mtlr2 and hkb-mtlr4), MDCK cells (ATCC® CCL-34™), HEK-Blue® Nulls (Invitrogen, Cat. # hkb-null1v and hkb-null2), NS0 murine myeloma cells (ECACC 85110503), BALB/c AB1-HA malignant mesothelioma derived from AB1 cells (ECACC 10092305) and the 4T1-HA mammary carcinoma cells derived from 4T1 cells (ATCC® CRL-2539™) were maintained at 37 °C in 5% $CO_2$ in DMEM medium (Gibco) supplemented with 2 mM L-glutamine, 100 U ml$^{-1}$ of penicillin and streptomycin (Invitrogen) and 10% heat-inactivated fetal bovine serum (FBS, Gibco). *Spodoptera frugiperda* Sf9 cells (Invitrogen, Cat. # LSB82501) were maintained at 27 °C in SF900-II SFM medium (Gibco, Cat. # 10902096). BMDCs were obtained by differentiation from BM precursors after culture for 8 days with granulocyte macrophage colony-stimulating factor (GM-CSF)[65]. Cells were determined to be > 90% CD11c$^+$ by flow cytometry. Spleen cells were cultured in plates containing RPMI 1640, 10% FBS, 100 U ml$^{-1}$ of penicillin, 100 µg ml$^{-1}$ of streptomycin, 0.25 µg ml$^{-1}$ of Fungizone® and 50 µM of 2-mercaptoethanol. *Entamoeba histolytica* strain HM-1: IMSS (ATCC® 30459), *Tetrahymena thermophila* strain SB210 (ATCC® 30007) and *Paramecium tetraurelia* strain d4-2 (ATCC® 30759) were cultured as previously reported[66,67]. Briefly, *E. histolytica* was cultured at 37 °C in TYI-S-33 supplemented with 10% adult bovine serum and a vitamin mix; *T. thermophila* at room temperature (RT) in PPYE 1X medium and *P. tetraurelia* at 27 °C in a culture medium consisting of a sterile wheat grass infusion inoculated with bacteria (*Klebsiella pneumoniae*) the day before use and supplemented with β-sitosterol.

**Virus strains**. A mouse-adapted variant strain (muH5N1) obtained from an avian H5N1 virus (A/crested eagle/Belgium/1/2004) was obtained by five passages by lung-to-lung, after that, the muH5N1 rapidly caused the death of naive mice and began to propagate stably in lungs[35].

The mouse-adapted influenza virus A/Puerto Rico/8/1934 (H1N1) (PR8) was cultured by inoculation into the allantoic cavity of embryonated chicken eggs and passaging by MDCK cells[68].

**Production of recombinant VSPs and HA**. Sf9 cells were used for the production of recombinant proteins by the Bac-to-Bac Baculovirus Expression System® (Invitrogen, Cat. # 10359016). The specific DNA fragments, containing the full-length VSP extracellular region and the His$_6$ protein purification tag at the carboxy-terminus were cloned into pFastBac1 and amplified in *E. coli* DH5α. The proteins were purified by immobilized metal affinity chromatography (IMAC) using the ÄKTA® pure chromatography apparatus. Protein analysis was carried out using sodium dodecyl sulfate polyacrylamide gel electrophoresis (SDS–PAGE) and the western blots were performed using an anti-His$_6$ antibody (Roche, Cat. # 04905318001, dilution 1/5000) and specific monoclonal antibodies (mAb G10/4 for VSPH7, dilution 1/1000; mAb 9B10 for VSP9B10, dilution 1/1000 and mAb 7F5 for VSP1267, dilution 1/1000)[12] and AP-conjugated anti-mouse immunoglobulins (Cappel, Cat. # 59294, dilution 1/2000). For the production of recombinant HA from the influenza A virus (A/Hong Kong/156/97), the specific DNA fragment containing the HA (GenBank: AAF02306.1) full extracellular domain and the His$_6$ protein purification tag at the C-terminus was cloned into the pFastBac1 plasmid and expressed and purified as the ΔVSPs.

**Mouse anti-HA (H5N1) and anti-VSP1267 monoclonal antibodies**. BALB/c mice were immunized intraperitoneally on days 0, 7, 14, and 21 with 25 µg of purified recombinant proteins, emulsified in Sigma Adjuvant System (Sigma, Cat. # S6322). Mice were boosted on day 28 intravenously with 10 µg of the protein. Three days later, mice were euthanized and the spleen cells used for fusion to NS0 myeloma cells. Polyethylene glycol was used as the fusing agent and the cells were incubated for 16 h. Subsequently, HAT (hypoxanthine, aminopterin, and thymidine) (Sigma, Cat. # H0262) was added to the medium[69]. The hybridomas were cloned by limiting dilution and the supernatants were screened by western blotting and indirect immunofluorescence with *G. lamblia* trophozoites (mAb 7F5, anti-VSP1267) or by western blotting and enzyme-linked immunosorbent assay (ELISA; mAb 15E4, anti-HA H5N1). MAbs were purified from supernatants screened using the ÄKTA® pure chromatography apparatus (HiTrapTM Protein G HP column, GE Cat. # 17-0405-03).

**Giardia lamblia cultures, lysates, and immunofluorescence**. *G. lamblia* WB and GS/M were cultured and cloned in the TYI-S-33 medium at pH 7.0 with 10% adult bovine serum and bovine bile (0.5 mg ml$^{-1}$) in anaerobiosis at 37 °C[11]. For lysates, trophozoites were sonicated in PBS with eight 30-s bursts and centrifuged at 10,000 × *g* for 20 min. The supernatant was collected and the protein concentration was determined using the Bradford method. For immunofluorescence, trophozoites were obtained and fixed by incubation in 100% cold methanol. Cells were labeled

with mAb 7F5-FITC (dilution 1/1000) for VSP1267[11,12]. DNA staining was performed with DAPI.

**Proteolytic assays**. Trophozoites from *G. lamblia*, *E. histolytica*, *T. thermophila*, and *P. tetraurelia*, and mouse myeloma cells NS0 were incubated during 90 min under different conditions. In the cases of *T. thermophila* and *P. tetraurelia*, PBS 0.1X was used to prevent osmotic stress. Cell viability was determined by simultaneous staining with Fluorescein Diacetate/Propidium Iodide[70]. For proteins, 1 µg of purified ΔVSP and HA, or 10 µg of *Giardia* lysate and VLP, were treated at 37 °C for 90 min with variable concentrations of trypsin (Sigma-Aldrich); different mouse intestinal (IE) or stomach (SE) extract dilutions; or at variable pHs. In addition, previous to trypsin digestion, 1 µg of purified ΔVSP1267 was subjected to different treatments: boiling (5 min, 100 °C); EDTA (2 mM, 15 min, RT); EDTA (2 mM, 15 min, RT) plus $CuSO_4$ or $ZnCl_2$ (10 mM, 15 min, RT); TCEP (5 mM, 15 min, RT); TCEP (5 mM, 15 min, RT) plus NEM (1.5 mM, 2 h, RT); NEM (1.5 mM, 2 h, RT). The reactions were stopped by adding a protease inhibitor cocktail (Complete®, EDTA-free, Roche), placed in sample buffer and boiled for 5 min. The IE and SE were obtained from normal BALB/c mice, fasted overnight. For SE, stomach was mechanically homogenized in 200 µl of ice-cold PBS pH 2, centrifuged at 10,000 × *g* for 15 min, and the supernatants were then used. For IE, the first portion of the small intestine (10 cm) was filled with 1 ml of PBS pH 7.4 and incubated at 4 °C for 30 min. The IE was recovered and centrifuged at 10,000 × *g*. Uncropped scans of western blots are provided in Supplementary Fig. 4.

**VLP expression plasmids**. For pGag-eYFP, the cDNA sequence encoding the Gag capsid protein of the MLV Gag (UniProt: P0DOG8.1) without its C-terminal Pol sequence was obtained by enzymatic digestion from plasmid pBL36-HCV[21] and cloned into the phCMV expression vector as a fusion protein with the enhanced yellow fluorescent protein (eYFP). For pHA and pNA, the cDNA sequences encoding HA and NA from the H5N1 influenza A/Hong Kong/156/97 (H5N1) virus (AF028709, AF046089, respectively) and the sequences encoding HA and NA from the H1N1 influenza strain A/Puerto Rico/8/34 (AGQ48050.1 and AGQ48041.1, respectively) were also cloned in the phCMV expression vector. All plasmids were verified by sequencing.

**Immunofluorescence and TEM of transfected HEK293 cells**. Transfected cells were harvested 48 h post-transfection and fixed by incubation in 100% cold methanol. Cells were labeled with the specific corresponding antibodies: mAb 7F5 for VSP1267, mAb 15E4 for HA-H5N1 (dilution 1/1000), rabbit pAb for NA-H5N1 (USBiologicals, Cat. # I7649-48, dilution 1/2000)) and mAb R187 for Gag (ATCC® CRL-1912™, dilution 1/2000). DNA staining was performed with DAPI. Images were taken using a Hamamatzu ORCA ER-II camera mounted on a Leica IRBE inverted fluorescence microscope (N.A. 1.40). TEM of cryosections was performed by applying the Tokuyasu technique and incubated with anti-HA or anti-VSP and anti-mouse coupled 5-nm gold particles (Sigma-Aldrich, Cat. # G7527, dilution 1/50). Finally, the samples were infiltrated with a mixture of 1.8% methylcellulose and 0.5% uranyl acetate for 5 min and air-dried. Images were obtained using a Zeiss LEO 906-E TEM.

**VLP generation, production, purification, and validation**. VLPs were produced by transient transfection of either HEK293 cells or HEK293-1267/H7/9B10 cells, with pGag, pHA and pNA plasmid DNA using PEI as transfection reagent. Cells were transfected at 70% confluence in T175 flasks with 70 µg of total DNA per flask at a PEI: DNA mass ratio of 3:1. VLP-containing supernatants were harvested 72 h post-transfection, filtered through a 0.45 µm pore size membrane, concentrated 20x in a centrifugal filter device (Centricon® Plus-70-100K, Millipore Cat. # UFC710008) and purified by ultracentrifugation through a 20% sucrose cushion in an SW41T Beckman rotor (25,000 rpm, 4 h, at 4 °C). Pellets were resuspended in sterile TNE buffer (50 mM Tris-HCl pH 7.4, 100 mM NaCl, 0.1 mM EDTA)[21,22]. Proteins were measured using the Bradford method. For western blotting proteins were resolved by 10% SDS–PAGE and transferred onto PVDF membranes before incubation with specific primary antibodies. Alkaline phosphatase-conjugated secondary antibodies were used and they were detected by BCIP/NBT substrate. Uncropped scans of western blots are provided in Supplementary Fig. 4. Direct immunofluorescence assay using 7F5-FITC was performed to confirm the presence of VSP onto VLPs. Hemagglutination assay was based on a previously described method[71] using a commercial flu vaccine as positive control. Briefly, two-fold serial dilutions of 50 µl of VLPs starting at 25 ng µl$^{-1}$ were used and 50 µl of a 0.5% chicken red blood cell suspension were added to each well, and the plates incubated for 2 h at RT. The HA titer was calculated as the highest dilution of VLPs agglutinating red blood cells. Additionally, VLP samples were prepared for TEM using the air-dried negative staining method. The samples were examined in a JEOL EXII 1200 electron microscope. Finally, the samples were analyzed using a Nanosight NS300 in a light scatter mode. The nanoparticle tracking analysis software (NTA 3.1) defined the concentration, size, and intensity of the particles within the samples. Each sample was analyzed in triplicate, and each replicate was measured twice.

**Dendritic cells analysis**. BMDCs were cultured with different stimuli for 48 h. Supernatants were collected and assayed for cytokine production using the CBA Mouse Inflammation Kit (IL-6, IL-10, IFN-γ, TNF-α, and IL-12p70) (BD Biosciences, Cat. # 552364), according to the manufacturer's instructions. Cells were stained with: CD11c-PE (eBioscience, Cat. #12-0114, dilution 1/50), CD86-APC (eBioscience, Cat. # 17-0862, dilution 1/250) and CD40-PerCP-eFluo710 (eBioscience, Cat. # 46-0401, dilution 1/50) and analyzed in an Accuri® C6 flow cytometer (BD). Gating strategy is shown in Supplementary Fig. 3a. Appropriate isotype controls were used.

**TLR reporter cells assays**. For the Quanti-Blue SEAP Reporter Assay (InvivoGen, Cat. # rep-qb1), HEK-Blue® TLRs cells were stimulated with the different compounds for 16 h, according to the manufacturer's instructions. Each reporter line was induced in parallel with a positive control sample. The HEK-Blue® Null2 and Null1-v were used as a negative control; the secretion of SEAP was induced with TNF-α. No activation of Null lines was observed, indicating the specificity of the reactions.

**VLP uptake/binding assay**. BMDCs were placed on ice for 30 min. A total of 50 μg ml$^{-1}$ eYFP fluorescent VLP-VSP/G or plain VLP were added to the cells. Sixty minutes later, the cells were washed to remove the unbound VLPs and placed either on ice or at 37 °C for 1 h. Finally, the cells were harvested and the fluorescence was analyzed by flow cytometry. Gating strategy is shown in Supplementary Fig. 3a. VLP uptake was quantified by measuring the signal of cells treated at 37 °C.

**Immunizations**. BALB/c mice were fasted 4 h and then orally immunized with four weekly doses of 100 μg of different VLPs. For subcutaneous immunization, four weekly doses of 10 μg of different VLPs were administered. Animals from the negative control group (naive) received oral immunizations with vehicle alone. Animals were not anesthetized during immunizations. The control groups received two doses of alum/antigen mixture: 10 μg recombinant HA (H5N1) or 10 μg of VLP-HA (H1N1) in 50 μl of PBS plus 50 μl Imject® Alum.

For the assays of pre-existing anti-VSPs antibodies, BALB/c mice were orally immunized with four oral weekly doses of 50 μg of recombinant ΔVSP1267. In parallel, another set of animals were administered with control PBS. One week after the last immunization, blood and fecal samples were collected to evaluate the presence of anti-ΔVSP1267 antibodies. These two groups were then immunized with VLP-HA/VSP-G (H5N1), according to the protocol previously described, and 1 week after the last dose blood and fecal samples were obtained to evaluate the presence of anti-HA antibodies.

**Splenocytes isolation and analysis**. Single cell suspensions from spleen were obtained and cultured on 96-well plates for 48 h with PBS, recombinant HA (1 μg ml$^{-1}$) or ConA (5 μg ml$^{-1}$). Cytokines in supernatants were measured using the CBA Mouse Th1/Th2/Th17 kit (BD Biosciences Cat. # 560485) according to the manufacturer's instructions.

**T-cell response**. The frequency of HA-specific T cells was analyzed by a standard IFN-γ ELISPOT assay (MabTech, Cat. # 3321-2 H). Splenocytes (5 × 10$^5$ cells per well) were stimulated with 1 μg ml$^{-1}$ of recombinant HA. PBS and Con A (5 μg ml$^{-1}$) were used as negative and positive controls, respectively. Spots were counted using the AID® ELISPOT reader.

**Fluid collection**. Blood was collected weekly from the retro-orbital sinus of mice and serum was separated. Fecal pellets were resuspended in PBS containing protease inhibitor (Complete® Protease Inhibitor Cocktail; Roche) and 0.1% sodium azide, at a ratio of 0.1 g per 500 μl. After 30 min on ice, the mixture was centrifuged at 10,000 × g and the supernatant was stored at −80 °C. BAL was collected through the trachea by injection-aspiration of 1 ml PBS with protease inhibitors.

**Enzyme-linked immunosorbent assay (ELISA) tests**. The levels of IgG, IgG$_1$, IgG$_{2a}$ or IgA antibodies against HA (H5N1 or H1N1) were determined by ELISA. The following horseradish peroxidase-conjugated antibodies were used: goat anti-mouse IgG$_1$ (SouthernBiotech, Cat. # 1070-05, dilution 1/2000); goat anti-mouse IgG$_2$ (SouthernBiotech, Cat. # 1080-05, dilution 1/2000); goat anti-mouse IgG (Molecular Probes, Cat. # G-21040, dilution 1/2000). For IgA, biotin anti-mouse IgA (BioLegend, Clone RMA-1, dilution 1/5000), and streptavidin-horseradish peroxidase-conjugate (BD Biosciences, Cat. # 51-9002208, dilution 1/1500) were used. The 3,3',5,5'-tetramethylbenzidine (TMB, BD Biosciences) was used to reveal the reaction. Optical density (OD) was measured on a microplate reader at 450 nm. To evaluate the presence of anti-VSP1267 IgG and IgA antibodies, ELISA plates were sensitized with recombinant ΔVSP1267 (10 μg ml$^{-1}$). Serum dilution 1/1000 and fecal dilution 1/2 were assayed.

**Antibody microneutralization assays**. Microneutralization assays using MDCK cells and 100 TCID$_{50}$ of A/Puerto Rico/8/1934 (H1N1) were performed. Briefly, serial dilutions of complement-inactivated serum were incubated with live virus at RT for 1 h and then added to monolayers of MDCK cells, which were incubated for a further 3 day period to examine the presence of cytopathic effect. The neutralizing titer was defined as the reciprocal of the highest dilution of serum at which the virus infectivity was completely neutralized in 50% of the wells[71].

**Cytotoxicity assay**. AB1-HA mesothelioma cells were used as target cells and labeled with 2 μM carboxyfluorescein succinimidyl ester (CFSE) followed by a 4 h incubation period with splenocytes from immunized mice (effector cells) at an E:T ratio of 20:1 in U-bottom 96-well culture plates at 37 °C[72]. The cells were harvested, stained with propidium iodide for 5 min and then subjected to flow cytometry analysis. Gating strategy is shown in Supplementary Fig. 3b. The death percentage was calculated on CFSE-positive cells. The mean percentage of each condition was calculated from three replicate wells.

**Virus protection assays**. Ten days after the last immunization, mice were anesthetized and intranasally challenged with 50 μl of a suspension containing 10$^2$ or 10$^4$ 50% lethal dose (LD$_{50}$) of mouse-adapted variant strain muH5N1 virus[20]. Mice were monitored daily for weight loss and survival during 2 weeks and those mice that exhibited over 25% body weight loss were sacrificed in accordance with the guidelines.

**Tumor protection assays**. Ten days after the immunization protocol, 1 × 10$^5$ AB1-HA mesothelioma cells were injected subcutaneously into the right flank of mice. Tumor-inoculated mice were sacrificed when average tumor diameters reached 15–20 mm. Tumor volume (mm$^3$) was determined using Vernier calipers ($L$ x $W^2$/2)[36]. For therapeutic vaccination, 20,000 4T1-HA cells were injected into the mammary gland or 1 × 10$^5$ AB1-HA cells were injected as described above into groups of BALB/c mice. Then, when the tumor was palpable, the animals were orally immunized with four doses of 100 μg of different VLPs or vehicle every 3 days.

**Statistical analyses**. For in vitro proteolytic densitometry experiments, two-tailed unpaired Student's $t$-test was used. Prism (GraphPad Software) was used to perform one-way or two-way ANOVA on datasets with Tukey's multiple comparisons test or Bonferroni post-test, respectively. Kaplan–Meier life survival curve analysis was performed using the log-rank Mantel-Cox method for curve comparison analysis. All figures show mean ± s.e.m. Statistically significant differences are indicated in each graph as $^*p < 0.05$, $^{**}p < 0.01$, and $^{***}p < 0.001$ and n.s. = not significant.

**Reporting summary**. Further information on experimental design is available in the Nature Research Reporting Summary linked to this article.

## Data availability
All relevant data are available from the authors. The source data underlying Figs. 1b–e, 2a–c, 3c, 4a–b, 4d–f, 5a–i, and 6a–h are provided as a source data file. The data associated with the paper has been deposited in a persistent repository with this identifier: https://doi.org/10.17605/OSF.IO/9WDCA [https://osf.io/9WDCA/].

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

## Acknowledgements

We thank Damian Peralta for production of recombinant VSPs, Sergio Oms for providing professional animal care, Carolina Leimgruber for performing electron microscopy, and Marcela F. Lopes for critical reading of the manuscript. This work was supported by grants from FONCYT (PICT-13469, PICT-2703, PICT-E 0234, and PICT-2116), CONICET (D4408), UCC (80020150200144CC) and MinCyT (Res. 204/11) of Argentina, a Georg Forster Award of the Alexander von Humboldt Foundation of Germany to H.D.L., and grants from the Institut National du Cancer (LSHB-CT-04-005246) and the European Union (EC-FP6-COMPUVAC) to D.K.

## Author contributions

M.C.S. and L.L.R. performed most the experiments, blinded or in parallel. C.G.P., P.G.C., and B.B. constructed the mammalian plasmids and performed preliminary experiments; E.P. performed and supervised preliminary experiments; R.A.M. constructed and validated the insect expression plasmids and generated the recombinant ΔVSPs; A.S. generated monoclonal antibodies; P.R.G., R.R.T., A.H.T., and J.P.P. performed VLP size determination and validation by electron microscopy; N.R.-V. and J.E. helped perform neutralizing antibodies assays; E.A.F. helped on the statistic analyses of the data; L.B. and T.S. suggested experiments and analyzed the data; D.K. and H.D.L. conceived the project and designed the experiments. M.C.S., L.L.R., D.K., and H.D.L. wrote the paper. All authors read and commented on the manuscript.

## Additional information

**Competing interests:** The authors declare no competing interests.

