## [Peer Review File · Nature Communications]

Reviewers' Comments:

Reviewer #1:

Remarks to the Author:

COMMENTS TO THE ARTICLE ENTITLED: "Efficient oral vaccination by bioengineering 1 virus-like particles with protozoan surface proteins" By Marianela C. Serradell, Lucía L. Rupil, Román A. Martino, César G. Prucca, Pedro G. Carranza, Alicia Saura, Elmer A. Fernández, Pablo R. Gargantini, Albano H. Tenaglia, Juan P. Petiti, Renata R. Tonelli, Nicolás Reinoso-Vizcaino, José Echenique, Luciana Berod, Eliane Piaggio, Bertrand Bellier, Tim Sparwasser, David Klatzmann and Hugo D. Luján

This work describes a novel strategy to develop oral vaccines based on the resistance of parasitic surface proteins containing CXXC motifs, with an emphasis in VSPs from *Giardia duodenalis*' surface. They report its protective role over heterologous proteins covering the surface of VSP-pseudotyped VLPs as well as its adjuvant function in the development of immune responses using HA as a model for testing antibody- and cellular-based responses against viral and HA-containing tumor challenges respectively. In general, the Results showed evidence regarding the native resistance of at least three giardial VSPs (9B10, 1267 and H7) to proteolysis, extreme pH and stability under different storage temperatures. Further when Influenza antigens were included in VSP-pseudotyped VLPs, they produce a remarkable immune response against the flu antigens. Oral vaccination with those VLPs protects mice from live Influenza virus challenges and from the development of tumors expressing the vaccinal antigen.

The work presented is well designed and it has been carried out in a good way hence the findings are experimentally supported. However, since the chimeric virus like particles have been decorated with VSPs and expressing model antigens, it will be important that the authors carried out experiments to validate the infectivity of VSP-(9B10, 1267 or H7)-expressing *Giardia* trophozoites and the patterns of anti-*Giardia* cytokines and antibodies elicited by the VLP-HA/VSP-G vaccine that proved to be efficient against HA-harboring viruses and tumoral cells. These data would give a broader notion on the utility of the vaccine and the possibility to test its anti-parasitic potential. This will support the use of the VLP-HA/VSP-G vaccine in other systems.

There are several works that have been reported regarding the use of Virus-like particles (VLPs) that represent a specific class of subunit vaccine that mimic the structure of authentic virus particles. In these it has been demonstrated that the VLPs are recognized readily by the immune system and present viral antigens in a suitable conformation to induce vigorous immune responses. Up to now, VLPs have been produced for more than 30 different viruses infecting humans and other animals and VLPs have also recently found application as scaffolds in nanoparticle biotechnology. Therefore, the authors need to discuss their results in the light of previously reported findings regarding the use of chimeric VLPs in inducing immunity to several viral proteins.

Further, since the authors mentioned that "the versatile oral vaccine platform based on VSP-pseudotyped VLPs can be easily adapted to different antigens from infectious agents or tumors and has the attributes to potentially help reduce reticence to vaccination, facilitate mass vaccination programs and be used in remote areas of the world where vaccine refrigeration is impractical". The authors need to discuss several aspects related to technical challenges for VSP-pseudotyped VLPs production. These include expression system in which the right balance between yields & VLPs composition should be considered; culture mode, transduction-related process parameters, the 'envirome' related parameters (e.g., dissolved oxygen concentration, pH, temperature, agitation rate, inlet gas flow and composition, volume or fluid dynamics) thermodynamics of VSP-pseudotyped VLPs production, formation (i.e., in vitro or in vivo). Also, other aspects that need to be discussed by the authors, and which are a major challenge in the preparation of these chimeric particles, are issues

related to the identification and overcome up- and downstream bioengineering aspects in the design of novel bioprocesses capable of delivering clinically viable VLP-derived bioproducts to the market in a timely manner and at a reduced cost.

Reviewer #2:

Remarks to the Author:

This is an interesting study. The concept of oral immunization by combining variant-specific surface protein (VSPs) and immunogens into VLPs is novel. However, some concerns must be addressed before acceptance for publication:

Major comments:

1. In Fig. 2, a recombinant protein expressed and purified from the same approach, like the recombinant HA in this study, should be tested for innate signaling activity as a control in these experiments. The negative TLR4/TLR2 activating activities of the control protein will help to exclude the possible contamination of TLR4 or TLR2 ligands in purified VSP, thus as LPS. In addition, the chimeric VLPs used for immunization should be included in this study as well because VLP-HA/VSP, not purified VSP, is what the authors wanted to study after all.
2. In Fig. 3, a less severe low pH treatment (like pH 5, which promotes HA to adapt to the post-fusion conformation but not degrade the proteins in a short time) followed with hemagglutination activity assays to VLP-HA, VLP-HA/VSP-G, and influenza viruses may give you clues if the HA in the VLP-HA/VSP is accessible to reagents. My understanding is that the HA is accessible because the chimeric VLPs can agglutinate red blood cells (S. Fig. 3c). This indicates HA is accessible to proteases (trypsin) as well. The proteolysis resistance of HA in VLP-HA/VSP-G, not in VLP-HA, must have a different mechanism underlying, not only the physical mask by VSP-G co-incorporated. What is it?
3. In vaccination and challenge studies (Fig.4 and Fig.5), a control of VLP-HA plus VLP-VSP-G (mixture) should be included to demonstrate if VSP must be physically associated with HA in the same VLP to exhibit adjuvant effect or protective effect to antigen HA. A soluble HA plus Alum is not a good control because the HA is in a membrane-anchored form in the VLPs. They are two different immunogens in comparison. VLP-HA plus Alum can be included as another control.
4. For live virus challenge versus anti-tumor challenge experiments, the protective immunological mechanisms can be very different. For anti-tumor immunity, T cell response (CTL), maybe ADCC as well, are major immunological effectors. For live virus challenge, high titers of neutralizing antibodies may be enough to confer protection. The authors may need to discuss the contribution of the two immune branches in triggering the anti-virus and anti-tumor protection seen in this study.

Minor comments;

1. In Fig5, only survivals were presented. A clinical mark, like the body weight change, upon challenge infection should be displayed. For example, mice lost 15%- 25% of body weight but survived are not considered as a good protection.
2. Lines 108-109 said 'These results demonstrate that efficiently targeting the gut-associated lymphoid tissue (GALT) of the upper small intestine...'. But no data indicate what GALT uptake antigens, also no evidence shows the antigen uptake process for the chimeric VLPs used in the study.
3. Lines 282: "however, VLPs have only been administered parentally.....", this is not true. VLP vaccines have been delivered by various routes, including intranasal, intravaginal, intrarectal, skin vaccination with microneedles.....

Reviewer #3:

Remarks to the Author:

This manuscript describes an oral immunization approach taking advantage of the surface protein component of an intestinal protozoan, *Giardia lamblia*, to safely deliver virus-like particle (VLP) vaccine. This is based on earlier research by this group showing that the cysteine-rich coat proteins of *G. lamblia* containing CXXC-rich motifs, called variant specific proteins (VSPs) that provide resistance from proteolytic degradation and low pH, both main concerns for oral vaccine antigen integrity. The authors have engineered nanoparticles murine leukemia virus gag based VLPs pseudotyped with the VSPs and the envelope protein of vesicular stomatitis virus and incorporating the HA protein of influenza virus (VLP-HA/VSP-G) as a model VLP vaccine for a proof-of-principle study. Data presented demonstrates resistance of the VSPs and VLPs w/, but not w/o VSPs. to proteolytic degradation by trypsin as well as mouse intestinal and stomach extracts in addition to resistance at pH values 2.0 and 10.0. It is shown that the VSPs as well as VSP-containing VLPs can activate innate immune system via TLR-4 signaling and thus can provide adjuvant function to the vaccine. The VSP-pseudotyped VLPs induced antigen-specific humoral and cellular immune responses, relatively better than native VLPs after oral delivery in mice that translated into protection against challenge with influenza virus or tumor cells expressing HA. Protection against influenza virus seemed to correlate with high neutralizing antibody response and T cell cytolytic activity. Therapeutic oral vaccination protected majority of mice injected with 4T1 breast cancer cells expressing the HA model antigen, AB1 mesothelioma cells expression HA. The reason attributed to the lack of protection against AB1 was the fast growing nature of these tumors. Overall, the manuscript describes a novel oral vaccination concept and the data presented is strong and convincing for the most part. The major concerns are:

1. Figure 4 shows data demonstrating oral immunization with VLP-HA/VSP-G, but not VLP-HA leads to induction of systemic and mucosal antibody responses in addition to systemic T cell responses. It is important to also include data using these two vaccines delivered by a systemic and/or another mucosal route to realize the relative strength of oral delivery for the VSP-coated VLP vaccines. This is especially important because the VLPs as vaccine vehicles are proven to be highly immunogenic and efficacious in the mouse models and therefore, it is useful to see whether degradation from oral delivery is the only reason for their failure to induce immunity.
 - a. Related to the above comment is the question of whether or not higher viral challenge dose or tumor cell burden would be also resisted by this vaccination. Such comparison either from new experiments or at least discussing in light of historical data would strengthen the significance of this study.
 - b. Therapeutic tumor vaccination data further extends this concern because the novel strategy to protect oral vaccines doesn't seem to be applicable to fast growing tumors, which is not clear whether the route, formulation or some other reasons related to tumor burden and timing of vaccination contributed to the failure. In fact, in the discussion (lines 329-333) it is stated based on the 4T1 breast cancer model data that their study "highlights not only the strength of the induced cytotoxic immune response, but also its fast development, which can efficiently be used to avoid metastasis of fast growing cancer cells after surgical tumor removal". In light of the failure of the vaccine to show protection in the mesothelioma model, this appears to be incorrect and also over interpretation of the limited data.
2. It is useful to also show post-challenge (with virus or tumor cells) the vaccine-induced antibody and/or T cell immune responses exhibit anamnestic increases and whether it correlates with protective efficacy.

RESPONSES TO REVIEWERS:**Reviewer #1 (Remarks to the Author):**

COMMENTS TO THE ARTICLE ENTITLED: “Efficient oral vaccination by bioengineering 1 virus-like particles with protozoan surface proteins” By Marianela C. Serradell, Lucía L. Rupil, Román A. Martino, César G. Prucca, Pedro G. Carranza, Alicia Saura, Elmer A. Fernández, Pablo R. Gargantini, Albano H. Tenaglia, Juan P. Petiti, Renata R. Tonelli, Nicolás Reinoso-Vizcaino, José Echenique, Luciana Berod, Eliane Piaggio, Bertrand Bellier, Tim Sparwasser, David Klatzmann and Hugo D. Luján

This work describes a novel strategy to develop oral vaccines based on the resistance of parasitic surface proteins containing CXXC motifs, with an emphasis in VSPs from *Giardia duodenalis*’ surface. They report its protective role over heterologous proteins covering the surface of VSP-pseudotyped VLPs as well as its adjuvant function in the development of immune responses using HA as a model for testing antibody- and cellular-based responses against viral and HA-containing tumor challenges respectively. In general, the Results showed evidence regarding the native resistance of at least three giardial VSPs (9B10, 1267 and H7) to proteolysis, extreme pH and stability under different storage temperatures. Further, when Influenza antigens were included in VSP-pseudotyped VLPs, they produce a remarkable immune response against the flu antigens. Oral vaccination with those VLPs protects mice from live Influenza virus challenges and from the development of tumors expressing the vaccinal antigen.

COMMENT 1: The work presented is well designed and it has been carried out in a good way hence the findings are experimentally supported. However, since the chimeric virus-like particles have been decorated with VSPs and expressing model antigens, it will be important that the authors carried out experiments to validate the infectivity of VSP-(9B10, 1267 or H7)-expressing *Giardia* trophozoites and the patterns of anti-*Giardia* cytokines and antibodies elicited by the VLP-HA/VSP-G vaccine that proved to be efficient against HA-harboring viruses and tumoral cells. These data would give a broader notion on the utility of the vaccine and the possibility to test its anti-parasitic potential. This will support the use of the VLP-HA/VSP-G vaccine in other systems.

RESPONSE: We first thank the Reviewer for her/his positive assessment of our work. We share her/his interest regarding the use of this novel oral vaccine platform against a variety of parasites. In fact, given the robust induction of humoral and cellular immune responses, this approach could be used to target selected antigens from different stages of parasites having complex life cycles including intra- and extracellular forms, just by including the antigens in the same VLP or by combining VLPs harboring different antigens in the same formulation.

In this work, however, our main focus was to demonstrate the exploitation of VSPs for the development of oral vaccines against heterologous antigens, no *Giardia* ones. For parasites undergoing antigenic variation of variable surface antigens (*Giardia*, *Trypanosoma*, *Plasmodium*, *Babesia*, among others) this approach will not be suitable given the number of potential surface antigen variants (from dozens to thousands, depending on the parasite’s antigen family). Therefore, the possible use of this vaccination system against *Giardia* as suggested by Reviewer #1 is not possible since this protozoan clearly undergoes antigenic variation by continuously switching its VSPs on the trophozoite surface (see Introduction). For this reason, vaccinating with a single

VSP would not be an applicable strategy. Our laboratory has already designed a vaccine against *Giardia* comprising the oral administration of the complete repertoire of VSPs obtained from transgenic trophozoites in which the enzymes involved in the process of regulating antigenic variation were silenced (Prucca *et al.*, *Nature* 2008). This oral vaccine has been administered with proven efficacy in gerbils (Rivero *et al.*, *Nature Medicine* 2010), and young and adult dogs and cats (Serradell *et al.*, *npgVaccines* 2016). Moreover, in a recent publication, we characterized the immune response induced by this anti-*Giardia* vaccine. This study was carried out in gerbils, the most reliable animal model of giardiasis since mice are not infected with *G. lamblia* (Serradell *et al.*, *Infect Immun* 2017). In that work, we showed that the anti-*Giardia* oral vaccine comprising the entire repertoire of purified VSPs induced local (S-IgA) and systemic (serum IgG) immune responses against VSPs, without signs of inflammation and with high expression levels of IL-17, IL-6, IL-4, and IL-5.

In the present work, determination of anti-VSP antibodies after oral administration of VLP-HA/VSP-G chimeric particles was carried out and specific levels of both serum IgG and fecal S-IgA were detected. This information has not been included in this manuscript since it was considered irrelevant given our previous reports. Instead, we have now included results of the use of our oral vaccine against Influenza in mice previously immunized with recombinant Δ VSP1267 to determine if previous immunizations against VSPs were beneficial or harmful for the administration of VSP1267-pseudotyped VLP-HA. As the results showed no evident differences, we did not incorporate these results in the original version of the manuscript. These results are now included as Fig. 5g.

COMMENT 2: There are several works that have been reported regarding the use of Virus-like particles (VLPs) that represent a specific class of subunit vaccine that mimics the structure of authentic virus particles. In these, it has been demonstrated that the VLPs are recognized readily by the immune system and present viral antigens in a suitable conformation to induce vigorous immune responses. Up to now, VLPs have been produced for more than 30 different viruses infecting humans and other animals and VLPs have also recently found application as scaffolds in nanoparticle biotechnology. Therefore, the authors need to discuss their results in the light of previously reported findings regarding the use of chimeric VLPs in inducing immunity to several viral proteins.

RESPONSE: In the Discussion Section of our manuscript, the use of different VLPs cores for expression of homologous and heterologous antigens for immunization has been cited. Nevertheless, based on Reviewer #1 and Reviewer #2 suggestions, the paragraph starting in Line 279 of the original version has been modified in the revised version of the Discussion as follows (Line 305 to Line 326):

"We investigated the use of VSPs in vaccine design using VLPs as a platform. The most widely used vaccines are based on attenuated or inactivated bacteria or viruses, emphasizing the importance of a particulate structure for proper immunization. Recombinant vaccines are safer than traditional vaccines, but are often less immunogenic and usually require multiple doses and effective adjuvants⁴³. VLPs used for homologous vaccination are a highly effective subunit vaccine that mimics the overall structure of virus particles and thus preserves the native antigenic conformation of the immunogenic proteins without containing infectious genetic material^{17, 18}. VLPs also make excellent carrier molecules for the delivery of heterologous antigens because their particulate structure is readily taken up by antigen-presenting cells, and is thus able to prime long-lasting CTL responses in addition to antibody responses^{44, 45}. In addition,

VLPs based on enveloped viruses, such as retrovirus-based VLPs, have the unique property of being able to express the envelope protein of heterologous viruses under their proper tertiary and even quaternary structures, thus ideal for the triggering Nabs¹⁸⁻²¹.

The ease of designing antigen presenting VLPs offers a promising approach for the industrial production of vaccines against many diseases. VLPs have been produced in a wide range of taxonomically and structurally distinct viruses, combining unique advantages in terms of safety and immunogenicity as exemplified by the current vaccines against the Human Papilloma Virus (HPV) and Hepatitis B virus (HBV)^{17, 18}. On the other hand, although several routes of administration have been used in different vaccination trials with VLPs, including nasal⁴⁶, intravaginal³³, rectal⁴⁷, skin⁴⁸ and oral⁴⁹ routes, VLP vaccines that are available on the market have only been delivered parenterally, showing the same disadvantages that any other vaccine given by injection¹⁸.”

COMMENT 3: Further, since the authors mentioned that “the versatile oral vaccine platform based on VSP-pseudotyped VLPs can be easily adapted to different antigens from infectious agents or tumors and has the attributes to potentially help reduce reticence to vaccination, facilitate mass vaccination programs and be used in remote areas of the world where vaccine refrigeration is impractical”. The authors need to discuss several aspects related to technical challenges for VSP-pseudotyped VLPs production. These include expression system in which the right balance between yields & VLPs composition should be considered; culture mode, transduction-related process parameters, the ‘envirome’ related parameters (e.g., dissolved oxygen concentration, pH, temperature, agitation rate, inlet gas flow and composition, volume or fluid dynamics) thermodynamics of VSP-pseudotyped VLPs production, formation (i.e., in vitro or in vivo). Also, other aspects that need to be discussed by the authors, and which are a major challenge in the preparation of these chimeric particles, are issues related to the identification and overcome up- and downstream bioengineering aspects in the design of novel bioprocesses capable of delivering clinically viable VLP-derived bioproducts to the market in a timely manner and at a reduced cost.

RESPONSE: Regarding the concerns raised by Reviewer #1, we can state that enveloped VLPs expressing the MLV-Gag have already been produced for clinical trials (VBI Vaccines Inc.; NASDAQ: VBIV). This company has demonstrated the ability to manufacture VLPs with yields and purity that are expected to be suitable for vaccine production at a commercial scale. Similar to the approach presented in this article, VBI’s VLPs are produced after transient transfection of cells with plasmids encoding the MLV-Gag and target surface or internal proteins of interest. Data obtained regarding the manufacturing of these VLPs can be found at <http://www.vbivaccines.com/wp-content/uploads/2015/05/MVADS-2015.v5.pdf>.

We have now added a sentence to the paragraph regarding VLP technology (Line 284 of the original version of the manuscript):

Line 329 to 331: “Retrovirus-based VLPs expressing heterologous antigens have already been produced for clinical trials, demonstrating the ability to be manufactured with yields and purity that are expected to be suitable for vaccine production at a commercial scale²¹.”

Reviewer #2 (Remarks to the Author):

This is an interesting study. The concept of oral immunization by combining variant-specific surface protein (VSPs) and immunogens into VLPs is novel.

We thank the Reviewer for her/his positive assessment of our work.

However, some concerns must be addressed before acceptance for publication:

Major comments:

COMMENT 1: In Fig. 2, a recombinant protein expressed and purified from the same approach, like the recombinant HA in this study, should be tested for innate signaling activity as a control in these experiments. The negative TLR4/TLR2 activating activities of the control protein will help to exclude the possible contamination of TLR4 or TLR2 ligands in purified VSP, thus as LPS. In addition, the chimeric VLPs used for immunization should be included in this study as well because VLP-HA/VSP, not purified VSP, is what the authors wanted to study after all.

RESPONSE: Although the contamination with LPS has been ruled out in a timely manner, either by its quantification through the Limulus test or by the use of the Polymyxin B inhibitor, the requested assay with recombinant HA has been now included on the revised version of the manuscript. The activation of the TLR reporter cell lines by recombinant HA expressed and purified using the same approach as Δ VSP was added as Fig. 2b. On the other hand, the activation of TLR4/2 reporter cells was assessed with VLPs (VLP-HA y VLP-HA/VSP-G) and the results were added as Fig. 4b. The following paragraph was added at Lines 150 and 182:

“Conversely, recombinant HA expressed and purified using the same approach did not show activation of any of the receptors (Fig. 2b).”

“As observed with the recombinant proteins, VSP-pseudotyped VLP-HA but not VLP-HA induced the activation of TLR-4 reporter cells (Fig. 4b).”

COMMENT 2: In Fig. 3, a less severe low pH treatment (like pH 5, which promotes HA to adapt to the post-fusion conformation but not degrade the proteins in a short time) followed with hemagglutination activity assays to VLP-HA, VLP-HA/VSP-G, and influenza viruses may give you clues if the HA in the VLP-HA/VSP is accessible to reagents. My understanding is that the HA is accessible because the chimeric VLPs can agglutinate red blood cells (S. Fig. 3c). This indicates HA is accessible to proteases (trypsin) as well. The proteolysis resistance of HA in VLP-HA/VSP-G, not in VLP-HA, must have a different mechanism underlying, not only the physical mask by VSP-G co-incorporated. What is it?

RESPONSE: First, we performed the hemagglutination assay with the VLPs treated at pH 5 as suggested and we observed that both VLPs were able to agglutinate red blood cells at this pH. For that reason, this result was not included in the revised manuscript. Supplementary Fig. 3 now was moved as Fig. 3 of the main article to support the following point-

Second, regarding the mechanism by which the VSP can protect a heterologous antigen, we can comment that:

- VSPs do not act as protease inhibitors, since the stomach and intestinal extracts contain a variety of serine-, carboxy- and endo-proteinases, with different catalytic mechanisms of action (already stated in the Discussion Section).

- VSPs do not act by shielding the antigens (HA and NA), at least not completely, due to the fact that HA retains the ability to hemagglutinate red blood cells and NA allows the release of VLPs from the cell surface, indicating that these molecules are accessible to their ligands/substrates. Additional evidence to support this hypothesis comes from the electron microscopy images showed in the original Supplementary Fig. 3, which show that both HA and VSP are detected on the surface of VLPs when using gold-labeled monoclonal antibodies. This indicates that VSPs do not form an “umbrella” on the VLPs, which would shield HA and NA.
 - Regarding the mechanism of protection, at this time we can only speculate that it could be due to the formation of inter- and intra-molecular interactions among these proteins, possibly through metal or cysteine bridges, which can hinder the action of proteolytic enzymes. Fig. 1e shows that the treatment with TCEP and EDTA turns the recombinant VSP more sensible to trypsin digestion, supporting this hypothesis.
 - Finally, we expect to reliably know whether our hypothesis is correct once the high-resolution structure of VSPs is determined (we have been unsuccessful so far). Regardless of the outcome, there seems to be a high probability that the CXXC-rich domain is an important player regarding the proteolytic properties of VSPs.
- In summary, we now added these sentences to the Discussion Line 338 to Line 351: “How VSPs protect heterologous antigens is still an open question. However, we have shown that (a) the heterologous antigens must be present on the same particles that express the VSP-G, (b) Na and HA are exposed on the surface of VLPs since NA is accessible to its substrate and HA retains its hemagglutinating activity, (c) HA and VSP are exposed because they are detected on the surface of VLPs by immunoelectron and immunofluorescence microscopy, and (d) VSPs do not act as protease inhibitors.

All these results suggest that the heterologous antigens are not directly shielded by VSP-G molecules on the surface of the VLPs. Therefore; we hypothesize that intra- and inter-molecular interactions among these proteins, likely involving metal and disulfide bridges, play crucial roles in antigen protection. Indeed, when VSPs are treated with metal chelators or reducing agents their protective properties are lost. Nevertheless, the completion of the high-resolution structures of Δ VSP and Δ VSP-Ag complexes, which has been elusive until now, will allow a better understanding of the mechanism by which VSPs interact and protect heterologous antigens from the action of proteolytic enzymes."

COMMENT 3: In vaccination and challenge studies (Fig.4 and Fig.5), a control of VLP-HA plus VLP-VSP-G (mixture) should be included to demonstrate if VSP must be physically associated with HA in the same VLP to exhibit adjuvant effect or protective effect to antigen HA.

RESPONSE: As suggested by Reviewer #2, we have now orally administered the mixture of VLPs to mice and analyzed the produced antibody levels. As expected, we found that the VLP-HA did not induce the production of antibodies against HA, probably because the particles were digested within the GIT. Similarly, we observed that when VLP-HA was administered together with the VLP-VSP-G (VLPs lacking HA) no antibodies were found in serum, indicating that the VSP must be present in the same particle as HA to provide protection from proteolytic enzymes. This result coincides with our hypothesis that VSPs form intra- and inter-molecule bonds among the proteins present in the envelope of the VLPs (VSP-G, HA, and NA), and this hinders the action of proteases. These results were added as Fig. 5e and the text was modified at Line 207 to Line 211.

“To test whether VSPs must be physically associated with HA on the same VLP, a mixture of VLP-HA and VLP-VSP/G was administered orally to mice and the induction of specific antibodies anti-HA was analyzed. When HA was in VLPs separated from VLP expressing VSP-G, no anti-HA antibodies were found in serum (Fig. 5e), indicating that VSP-G must be in the same particle that HA to provide protection to the heterologous antigen.”

COMMENT 4: A soluble HA plus Alum is not a good control because the HA is in a membrane-anchored form in the VLPs. They are two different immunogens in comparison. VLP-HA plus Alum can be included as another control.

RESPONSE: We agree with the remark. In fact, most of the experiments were performed with VLP-HA plus Alum and only the challenge experiment with the influenza virus was done with the recombinant HA, which is a reference standard control in challenge experiments.

COMMENT 5: For live virus challenge versus anti-tumor challenge experiments, the protective immunological mechanisms can be very different. For anti-tumor immunity, T cell response (CTL), maybe ADCC as well, are major immunological effectors. For live virus challenge, high titers of neutralizing antibodies may be enough to confer protection. The authors may need to discuss the contribution of the two immune branches in triggering the anti-virus and anti-tumor protection seen in this study.

RESPONSE: Taking into consideration the suggestion of Reviewer #2, the Discussion Section starting at original Line 319 has been modified as requested in Lines 375-389: “Most importantly, these cellular and humoral immune responses translated into vaccine efficiency. We used two prototypic models for assaying humoral and cellular responses, in which the protective immunological mechanisms are essentially very different: protection from live virus infection and killing of tumor cells.

It is widely accepted that an effective anti-influenza vaccine must elicit NABs. Here, the protection from an intranasal challenge with live Influenza virus was indeed correlated with the generation of neutralizing anti-HA antibodies. As the formation of these antibodies is CD4⁺ helper T-cell dependent, these results also indicate that efficient immune cellular responses were triggered by orally administered VSP-pseudotyped-VLPs⁵⁹. On the other hand, protection from tumor challenge is largely dependent on the generation of cytotoxic T lymphocytes (CTL), with a possible contribution of antibody-dependent cell-mediated cytotoxicity and the induction of IFN- γ . Here, the protection from AB1 HA-expressing tumor cells observed after oral immunization with the VSP-pseudotyped-VLPs indicates that efficient cytotoxic responses were mounted. A positive correlation was also noted between IFN- γ production and the tumor growth control (Fig. 6d, f) while no such correlation was observed with antibody levels (Fig. 5a and 6d).”

Minor comments;

COMMENT 6: In Fig. 5, only survivals were presented. A clinical mark, like the body weight change, upon challenge infection should be displayed. For example, mice lost 15%- 25% of body weight but survived are not considered as a good protection.

RESPONSE: The requested figure has been added as Fig. 6b. And the following text was added at Lines 242-247:

“Challenged mice were monitored daily for disease signs (ruffled fur, dyspnea, lethargy) and body weight changes for 2 weeks (Fig. 6b). No significant body weight loss or clinical signs of infection were observed in non-challenged mice or in mice

vaccinated with oral VLP-HA/VSP-G. In contrast, unvaccinated mice, mice orally vaccinated with VLP-HA and subcutaneously vaccinated with recombinant HA plus Alum rapidly lost weight and died or had to be euthanized at 4 to 6 days post challenge due to severe clinical symptoms.”

COMMENT 7: Lines 108-109 say ‘These results demonstrate that efficiently targeting the gut-associated lymphoid tissue (GALT) of the upper small intestine...’. But no data indicate what GALT uptake antigens; also no evidence shows the antigen uptake process for the chimeric VLPs used in the study.

RESPONSE: Taking into account this observation and to better explain our conclusions and hypothesis, we have changed the content of the paragraph in the Discussion section (previous Line 108 and 334, respectively). Now Lines 402-408: “In the intestine, induction and regulation of mucosal immunity take place primarily in gut-associated lymphoid tissue (GALT)⁶⁰. *G. lamblia* colonizes the upper small intestine where it releases VSPs that are known to interact with the gut epithelial and immune cells⁶¹. Our results propose that a marked difference takes place between the large intestine and the upper small intestine in determining the outcome of the immune response. Thus, our results would suggest that efficiently targeting the GALT of the upper small intestine, instead of the tolerogenic large intestine, can generate protective immunity rather than tolerance.”

COMMENT 8: Lines 282: “however, VLPs have only been administered parentally.....”, this is not true. VLP vaccines have been delivered by various routes, including intranasal, intravaginal, intrarectal, skin vaccination with microneedles.....

RESPONSE: Considering this comment and the suggestions from Reviewer #1, the paragraph from Line 279 has been expanded in the new version of the Discussion (see response to Reviewer #1, Comment 2).

Reviewer #3 (Remarks to the Author):

This manuscript describes an oral immunization approach taking advantage of the surface protein component of an intestinal protozoan, *Giardia lamblia*, to safely deliver virus-like particle (VLP) vaccine. This is based on earlier research by this group showing that the cysteine-rich coat proteins of *G. lamblia* containing CXXC-rich motifs, called variant specific proteins (VSPs) that provide resistance from proteolytic degradation and low pH, both main concerns for oral vaccine antigen integrity. The authors have engineered nanoparticles murine leukemia virus gag based VLPs pseudotyped with the VSPs and the envelope protein of vesicular stomatitis virus and incorporating the HA protein of influenza virus (VLP-HA/VSP-G) as a model VLP vaccine for a proof-of-principle study. Data presented demonstrates resistance of the VSPs and VLPs w/, but not w/o VSPs. to proteolytic degradation by trypsin as well as mouse intestinal and stomach extracts in addition to resistance at pH values 2.0 and 10.0. It is shown that the VSPs, as well as VSP-containing VLPs, can activate the innate immune system via TLR-4 signaling and thus can provide adjuvant function to the vaccine. The VSP-pseudotyped VLPs induced antigen-specific humoral and cellular immune responses, relatively better than native VLPs after oral delivery in mice that translated into protection against challenge with influenza virus or tumor cells expressing HA. Protection against influenza virus seemed to correlate with high neutralizing antibody response and T cell cytolytic activity. Therapeutic oral vaccination protected the majority of mice injected with 4T1 breast cancer cells expressing the HA model antigen, AB1 mesothelioma cells expression HA. The reason

attributed to the lack of protection against AB1 was the fast-growing nature of these tumors. Overall, the manuscript describes a novel oral vaccination concept and the data presented is strong and convincing for the most part.

We first thank the Reviewer for her/his positive assessment of our work.

The major concerns are:

COMMENT 1: Figure 4 shows data demonstrating oral immunization with VLP-HA/VSP-G, but not VLP-HA leads to induction of systemic and mucosal antibody responses in addition to systemic T cell responses. It is important to also include data using these two vaccines delivered by a systemic and/or another mucosal route to realizing the relative strength of oral delivery for the VSP-coated VLP vaccines. This is especially important because the VLPs as vaccine vehicles are proven to be highly immunogenic and efficacious in the mouse models and therefore, it is useful to see whether degradation from oral delivery is the only reason for their failure to induce immunity.

RESPONSE: As suggested, VLPs were now administered by a systemic route (subcutaneously). This was now included as Fig. 5f. We observed that all particles produced anti-HA IgG antibodies. However, a high-level anti-HA S-IgA was only generated by the VSP-pseudotyped particles orally administered. The text was modified at Lines 212-219:

“Given that parentally administered VLPs are highly immunogenic^{19,33}, we also evaluated the effect of VSP using this route of administration. Similar to oral administration, subcutaneously administered VLP-HA induced specific anti-HA serum antibodies. However, subcutaneous immunization with VLP-HA/VSP-G achieved higher levels of anti-HA IgG than with VLP-HA. This indicates that the presence of VSP onto the VLP not only fulfills the protective effect observed using oral administration but also increases the immunogenicity of systemically administered VLPs (Fig. 5f). Conversely, a significant amount of fecal anti-HA IgA was only observed with oral VLP-HA/VSP-G vaccination (Fig. 5f).”

COMMENT 2: Related to the above comment is the question of whether or not higher viral challenge dose or tumor cell burden would be also resisted by this vaccination. Such comparison either from new experiments or at least discussing in light of historical data would strengthen the significance of this study.

RESPONSE: This is an interesting question. In this work, the viral load and tumor cells used to challenge assays were selected according to previous works of our group and by others. For now, we are asked to minimize the number of mice used since this is a Proof of Concept study. However, the Discussion section now included several sentences indicating the importance of this issue.

COMMENT 3: Therapeutic tumor vaccination data further extends this concern because the novel strategy to protect oral vaccines doesn't seem to be applicable to fast-growing tumors, which is not clear whether the route, formulation or some other reasons related to tumor burden and timing of vaccination contributed to the failure. In fact, in the discussion (lines 329-333) it is stated based on the 4T1 breast cancer model data that their study "highlights not only the strength of the induced cytotoxic immune response but also its fast development, which can efficiently be used to avoid metastasis of fast-growing cancer cells after surgical tumor removal". In light of the failure of the vaccine

to show protection in the mesothelioma model, this appears to be incorrect and also over interpretation of the limited data.

RESPONSE: These tumor cell lines grow extraordinarily fast (even the 4T1) when compared to human tumors. These are just limits of mouse models. However, from the comments, we have reformulated the paragraph including some of these observations (Original Line 228). The following text was modified at Lines 390-401:

"VSP-based oral immunization against tumors could possibly be used therapeutically. Although therapeutic oral vaccination was not effective against the mesothelioma tumor, it led to tumor eradication when using the HA-expressing 4T1 murine breast cancer. Several factors could contribute to the observed differential outcome. Among these, the anti-tumor immune response, tumor cell number, growth rate, size, and the particular tumoral microenvironment are likely the most significant³⁶. Nevertheless, the 4T1 tumor control/reversion highlights the strength, quality and the effectiveness of the induced cytotoxic immune response elicited by VLP-HA/VSP-G vaccine. It is well known that tumor fate is the uncertain result of a battle between tumor and immune cells in a particular environment³⁶. Nevertheless, the unexpected differences between these two HA-expressing tumors make them valuable starting models to better understand the molecular mechanisms underlying the observed effects and the potential manipulation of this vaccine platform to enhance its therapeutic potential against a variety of tumors^{36,37}."

COMMENT 4: It is useful to also show post-challenge (with virus or tumor cells) the vaccine-induced antibody and/or T cell immune responses exhibit anamnestic increases and whether it correlates with protective efficacy.

RESPONSE: The IFN- γ production showed in Fig. 5f (now Fig. 6f) was measured in the animals that had already developed tumors. In this case, a positive correlation was noted between IFN- γ production and tumor growth control (Fig. 6d). In contrast, no correlation was observed between antibody levels and tumor weight (Fig. 5a). The following text was added at Lines 387-389: "A positive correlation was also noted between IFN- γ production and the tumor growth control (Fig. 6d, f) while no such correlation was observed with antibody levels (Fig. 5a and 6d)."

We agree with the reviewer that more thorough immune investigations are now sought for designing future optimal vaccine efficacy, which we think fall behind the scope of this proof of concept study.

Notes:

- The long-term titer of serum antibody levels has been included as Fig. 5d.
- Supplementary Fig. 3 was moved as Fig. 3 of the main article.
- Additional minor modifications are indicated in the text of the file named "Revised Article with Changes Highlighted".

Reviewers' Comments:

Reviewer #1:

Remarks to the Author:

In their reply, the authors have addressed basically all to the comments raised by the reviewers. The revised version of the manuscript includes the new results and the authors have added the information requested by the reviewers. In general, the results and modifications added to the manuscript based on the comments made by the reviewers have indeed improved the contents of the work described in the revised version of the manuscript.

Reviewer #2:

Remarks to the Author:

The comments to the previous submission have been addressed sufficiently.

Reviewer #3:

Remarks to the Author:

The authors have adequately addressed the comments and I don't have any further questions/comments.

RESPONSE TO REVIEWERS:

REVIEWERS' COMMENTS:

Reviewer #1 (Remarks to the Author):

In their reply, the authors have addressed basically all to the comments raised by the reviewers. The revised version of the manuscript includes the new results and the authors have added the information requested by the reviewers. In general, the results and modifications added to the manuscript based on the comments made by the reviewers have indeed improved the contents of the work described in the revised version of the manuscript.

Reviewer #2 (Remarks to the Author):

The comments to the previous submission have been addressed sufficiently.

Reviewer #3 (Remarks to the Author):

The authors have adequately addressed the comments and I don't have any further questions/comments.

RESPONSE TO REVIEWERS:

We very much appreciate your assistance and comments for improving the quality of our work.